# Fertilization-induced synergid cell death by RALF12-triggered ROS production and ethylene signaling

Junyi Chen [1,3] ✉, Huan Wang [1,3], Jinlin Wang [1], Xixi Zheng [2], Wantong Qu [1], Huijian Fang [1], Shuang Wang [1], Le He [1], Shuang Hao [1] & Thomas Dresselhaus [2] ✉

Fertilization-dependent elimination of the persistent synergid cell is essential to block supernumerary pollen tubes and thus to avoid polyspermy in flowering plants. Little is known about the molecular mechanisms ensuring timely induction and execution of synergid cell death. We analyzed manually isolated maize synergid cells along their degeneration and show that they are gland cells expressing batteries of genes encoding small secreted proteins under control of the MYB98 transcription factor. This network is down-regulated after fertilization, while genes involved in reactive oxygen species (ROS) production, ethylene biosynthesis and response, senescence, and oxidative stress regulation are induced before synergid elimination and its ultimate fusion with the endosperm. We further show that the fertilization-induced RALF12 peptide specifically triggers mitochondrial ROS and apoptosis, while ethylene promotes synergid degeneration. In conclusion, this study sheds light on developmental programmed cell death (dPCD) in plants and provides a unique resource to discover unknown PCD regulators.

Programmed cell death (PCD) is a widespread phenomenon that is essential for all multicellular organisms. It occurs in a highly controlled manner to selectively remove damaged or no longer needed cells that are sacrificed for the good of the whole organism[1–3]. PCD in plants is generally classified into environmental PCD (ePCD), which is triggered by external biotic or abiotic factors, and developmental PCD (dPCD), which occurs during vegetative and reproductive development[3]. dPCD is particularly frequent in plant reproduction[4] and thus is crucial for fertilization success and seed production. During sex determination in many flowering plant species, carpel primordia cells are eliminated in male flowers, while stamen abortion occurs in female flowers. During microsporogenesis and megasporogenesis, tapetum cells and three of four megaspores are degenerated, respectively. PCD further occurs during nucellus and endosperm development but appears to be especially important for fertilization. During self-incompatible pollen-

pistil interactions, PCD prevents alien and self-pollen tubes from growing toward ovules. Upon compatible pollen tube reaching the female gametophyte, the pollen tube cell and two synergid cells undergo degeneration in a highly regulated two-step process[5–7]. $Ca^{2+}$ spiking in synergid cells and ROS-induced $Ca^{2+}$ channel activation in the pollen tube lead to simultaneous bursts of the pollen tube and one of the two synergid cells (i.e., receptive synergid), resulting in sperm cell release for double fertilization[8–10]. The second synergid cell (i.e., persistent synergid) remains intact during this step and maintains the capacity to attract additional pollen tubes in the case of fertilization failure[7,11,12].

In animals, it was shown that polyspermy (fusion of the egg by multiple sperm) leads to lethal genome imbalance and chromosome segregation defects during embryo development[13]. In flowering plants, the attraction of supernumerary pollen tubes is avoided to prevent

[1]Key Laboratory of Pesticide & Chemical Biology of Ministry of Education, Hubei Key Laboratory of Genetic Regulation and Integrative Biology, School of Life Sciences, Central China Normal University, Wuhan, Hubei Province, China. [2]Cell Biology and Plant Biochemistry, University of Regensburg, Regensburg, Germany. [3]These authors contributed equally: Junyi Chen, Huan Wang. ✉e-mail: junyi.chen@ccnu.edu.cn; thomas.dresselhaus@ur.de

polyspermy and to ensure chromosomal balance and progeny health[14]. Once fertilization is achieved, the persistent synergid is removed in a second step with some delay to block supernumerary pollen tubes and thus prevent polyspermy. Ultimately, the persistent synergid cell in the model plant *Arabidopsis thaliana* (*Arabidopsis*) was shown to fuse with the large fertilized central cell and is thus eliminated[15]. It remained unclear whether removal of the persistent synergid cell is achieved by PCD[4].

Despite its abundance and importance for plant life, the gene regulatory networks (GRNs) controlling the (i) induction (i.e., cell receiving signals that trigger the apoptotic pathway), (ii) effector (i.e., activation of lytic enzymes for cellular components cleavage), and (iii) degradation (i.e., cell disintegration and elimination) phases of dPCD are still poorly understood. This is mainly attributed to the occurrence of dPCD in embedded cells, which are surrounded by many other cells and thus are not accessible. Additionally, the initiation of dPCD is difficult to predict. During double fertilization in flowering plants, timely degeneration of synergid cells is crucial for reproductive success[5,6,15–19]. It has been shown that upon pollen tube arrival at the female gametophyte, RALF peptides released from the pollen tube interact with synergid-located FERONIA (FER), ANJEA (ANJ), and HER-CULES RECEPTOR KINASE1 (HERK1) receptor-like kinases to control pollen tube reception and rupture[14]. FER mediates high ROS levels at the filiform apparatus area, which mediates pollen tube rupture and sperm cell release for fertilization[10]. MITOGEN-ACTIVATED PROTEIN KINASE 4 (MPK4) prevents premature synergid cell death and FER-controlled ROS accumulation[6]. Besides, a 'cell death module' consisting of REM transcription factors VALKYRIE (VAL) and VERDANDI (VDD), both targets of the ovule identity MADS-box complex SEED-STICK-SEPALLATA3, controls the death of the receptive synergid cell[20]. The plant hormone ethylene has been implicated in persistent synergid degeneration as the persistent synergid maintains its integrity longer in the ethylene-insensitive mutant *ein3 eil1*[18]. However, mutants incapable of producing ethylene do not show a PCD-related phenotype in synergid cells[19] indicating that the ethylene signaling pathway, but not the hormone itself, might be involved in synergid cell death in *Arabidopsis*. It was suggested that an unknown ethylene-independent signaling pathway may control the death of the persistent synergid cell individually or synergistically with the ethylene-signaling pathway[19]. Nevertheless, the timely activation and execution of synergid, especially persistent synergid degeneration, and thus a fundamental principle in ensuring reproductive success remained unclear.

In this work, by overcoming a number of technical limitations, we developed methods to isolate maize synergid cells at defined stages, then systematically explored the characteristic degeneration processes and the molecular machinery responsible for driving this system. We determined the exact timing of the loss of synergid-specific GRNs and the induction of synergid PCD. PCD inducers, transcription factors, executors, as well as autophagy activities are strongly activated a few hours before synergid-endosperm fusion (SE fusion). Moreover, we show that the fertilization-induced autocrine signal ligand ZmRALF12 specifically triggers mitochondrial ROS maxima at the filiform apparatus adjacent region (FAAR) to induce dPCD. This PCD pathway works synergistically with activated ethylene production and signaling in sensitized synergid cells ensuring timely dPCD execution and corpse clearance shortly before SE fusion. In conclusion, this study uncovers a fundamental mechanism of dPCD during the essential fertilization process and provides a valuable resource for further studies investigating key molecular players responsible for driving dPCD in plants.

## Results

### Timing of persistent synergid degeneration and generation of stage-specific RNA-seq data

To investigate the degeneration of the persistent synergid cell after fertilization in maize, we first monitored the morphological changes of synergid cells using a synergid cell marker line *pZmES4::ZmES4-GFP* (Fig. 1a–d)[21] and additionally analyzed the inbred line B73 (Supplementary Fig. 1a–d). Before pollination, two synergid cells with comparable morphology were observed at the micropylar region of the female gametophyte (Fig. 1a and Supplementary Fig. 1a). 12 h after pollination (HAP), a dark and degenerated receptive synergid cell was visible, indicating that fertilization occurred (Supplementary Fig. 1b). The second synergid cell, termed the persistent synergid cell, was visible until 24-26 HAP (Fig. 1b, c). At about 25–28 HAP, the persistent synergid cell became abruptly invisible, leaving the degenerated receptive synergid and the zygote tightly attached at the micropylar region of the embryo sac (Supplementary Fig. 1c, d). Similarly, the synergid marker was no longer detectable at 28 HAP (Fig. 1d). These observations indicate that the persistent synergid cell has been completely removed at around 25–26 HAP, which corresponds to 17–18 h after fertilization (HAF), taking into consideration that it takes ~8 h for pollen tubes to germinate and grow towards the female gametophyte at a silk length of about 10 cm[22] (Fig. 1e–h).

Next, we developed methods to isolate living synergid cells and collected cells at defined stages after pollination/fertilization. Since many vacuoles were present in the peripheral region of the egg cells, they could be easily distinguished from synergid cells (Fig. 1i). Morphological changes of receptive synergid cells (they appeared dark after receiving pollen tube contents) facilitated discrimination of synergid cells before fertilization and persistent synergid cells shortly after fertilization (Fig. 1j and Supplementary Fig. 1b). Thereafter, synergid cells before pollination (SC), persistent synergid cells shortly after fertilization (pSC12, 12 HAP/about 4 HAF), during cell degeneration (pSC18, 18 HAP/about 10 HAF) and shortly before cell elimination (pSC24, 24 HAP/about 16 HAF, i.e., only ~1–2 h before cell deletion) were manually isolated (see Fig. 1i–l for isolated cells) and collected to generate high-timing-precision degenerating stage-specific transcriptomic data at cellular resolution. Three independent biological replicates were prepared for each cell stage and sequenced at a depth of over 53 million reads per library (Supplementary Data 1). A comparison of the three biological replicates within the same stage showed that the expression values between them were highly correlated (average $R^2 = 0.93$). Principal component analysis (PCA) further separated the synergid cell samples into four groups (Supplementary Fig. 2). In total, transcripts of 14,700–15,800 unique genes were detected in synergid cells at sequential developmental stages (Supplementary Data 1 and 2).

For the initial analysis of dynamic gene expression during the synergid cell degeneration processes, we focused on known synergid-expressed genes (Fig. 1m). *ZmES4*[21] transcript levels remain high after fertilization which correlated well with the ZmES4-GFP reporter activity shown in Fig. 1a–c. The *Arabidopsis* receptor-like protein kinase FERONIA and its coreceptor LORELEI were shown to act specifically at the synergid cell surface for pollen tube reception[23,24]. Their orthologs in maize, *ZmFERL1*, and *ZmLLG3*[25], were expressed in synergid cells and showed distinct activation patterns during synergid cell degeneration. We also found that *ZmMYB98*, the most highly expressed transcription factor gene in maize synergid cells, exhibited a substantial and continuous decrease during synergid degeneration (Fig. 1m). In summary, the observed gene expression patterns in synergid cells are in line with previous reports and expectations, which together with strong correlation between independent biological replicates assure the high quality and reliability of the data.

To gain further insight into the timing and scale of PCD triggering, activation, and execution in the persistent synergid cell, we identified significant upregulation of 596 genes in pSC12, 247 genes in pSC18, and 1506 genes in pSC24 (Fig. 1n and Supplementary Fig. 3). In total, we observed significant upregulation of 2727 genes in persistent synergid cells at 24 HAP compared to synergid cells. Substantial changes were observed between the transcriptional profiles of pSC18 and pSC24,

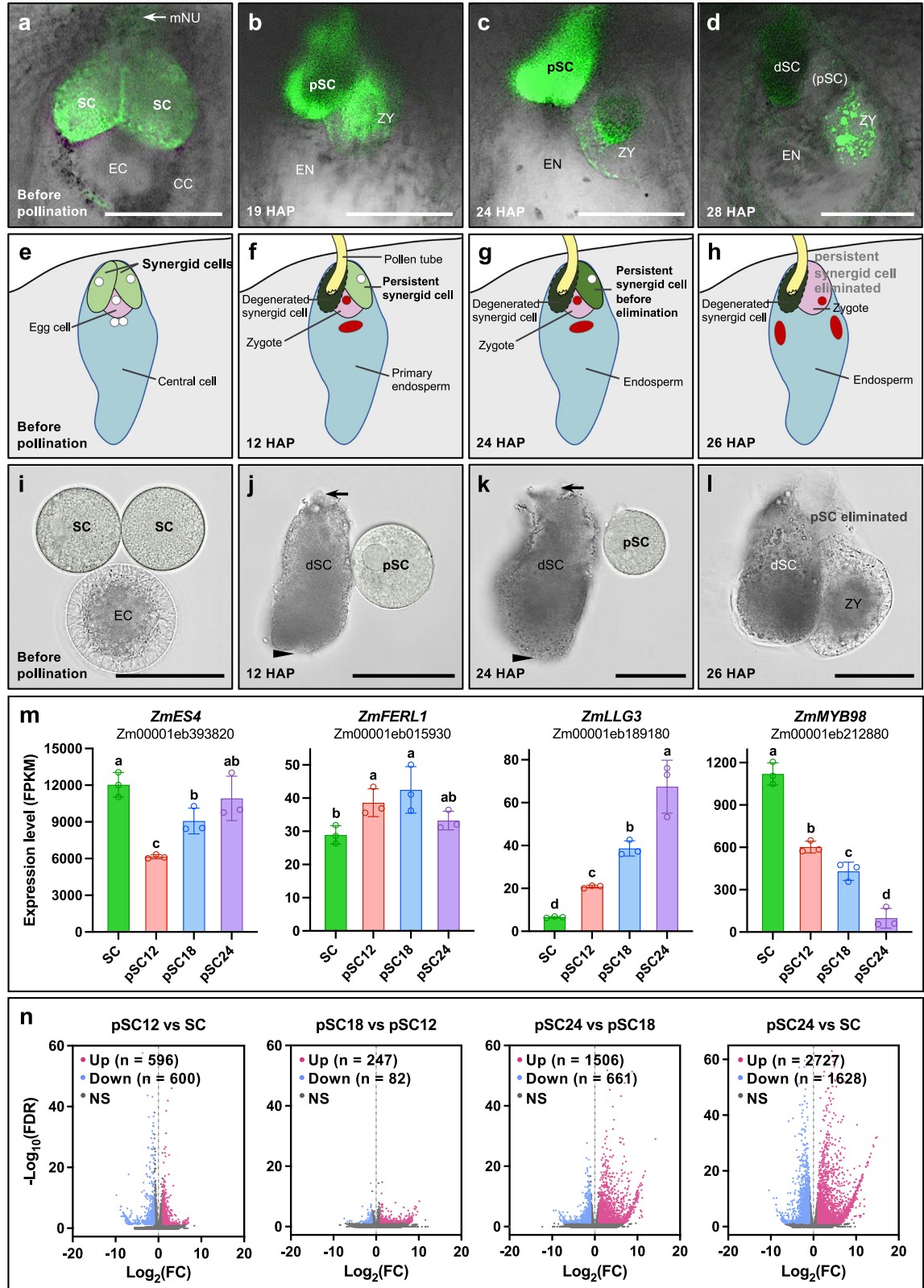

indicating that the major PCD-related activation wave occurs later after fertilization and only within a few hours before persistent synergid elimination.

## Cell-specific gene expression atlas in synergid and egg cells

Next, we analyzed gene expression patterns in synergid cells of maize before pollination. Among the top 30 genes with the highest expression values, 26 genes (86.7%) encode proteins with predicted N-terminal signal peptides (SPs), indicating that they are targeted to the secretory pathway. In contrast, in the neighboring egg cells/zygotes[22], only 10% of the top 30 genes encode proteins carrying SPs for protein secretion (Fig. 2a and Supplementary Data 3). Remarkably, in the TOP200 category (i.e., top 200 most highly expressed genes), substantially more synergid cell genes encode ER/Golgi localized

**Fig. 1 | Temporal resolution of transcriptome changes during synergid degeneration in maize. a–d** Tracing synergid cell degeneration using *pZmES4::ZmES4-GFP* as a marker at indicated times before and after pollination. *ZmES4* is expressed in synergid cells and weakly in the egg cell and zygote, respectively. At 28 HAP the synergid signal disappeared. **e–h** Scheme showing the timing of the double fertilization process and elimination of the two synergid cells. **i–l** Manually isolated synergid cells at indicated stages. At 26 HAP the persistent synergid cell is no longer visible and the zygote is tightly attached to the degenerated receptive synergid cell (**l**). Black arrows and arrowheads indicate pollen tube entrance and sperm cell release regions of the receptive synergid cell, respectively. **m** Expression pattern of known synergid-expressed genes at indicated synergid stages. Transcript levels are shown as FPKM (Fragments Per Kilobase of exon model per

Million mapped fragments) values (means ± SD) of three biological replicates. For statistical analysis, count data at the gene level were analyzed with DESeq2[64]. Stage-to-stage comparisons were performed and corrected for multiple testing over all genes and cell stage comparisons using false discovery rate (FDR). Letters above bars indicate significant differences (adjusted *P* < 0.05). Source data are provided in the source data file. **n** Volcano plots depict transcriptional dynamics of differentially expressed genes (DEGs) between indicated stages. Observation of GFP signals in *pZmES4::ZmES4-GFP* (**a–d**) and synergid cell isolation/observation at sequential stages (**i–l**) were repeated three times with similar results. CC central cell, dSC degenerated receptive synergid cell, EC egg cell, EN endosperm, mNU micropylar nucellus, pSC persistent synergid cell, SC Synergid cell, ZY zygote. Scale bars, 50 μm.

proteins (11.5% in synergid cells versus 1.5% in egg cells; Fig. 2b and Supplementary Data 3). Conversely, egg cells display remarkably enriched transcripts for mitochondrial proteins (Fig. 2b), indicating elevated mitochondrial biogenesis and/or activity in the female gamete. This result is consistent with the concept that maternal mitochondrial function is important for egg cell maturation and embryogenesis initiation[26] in flowering plants. Besides, egg cells and zygotes contain comparable highly abundant ribosomal transcripts, which are orders of magnitude higher compared with synergid cells (Fig. 2b). This is consistent with previous studies in animal species showing that large amounts of ribosomes/ribosomal transcripts are produced and stored in the female gamete before fertilization, to fulfil efficient protein production during embryogenesis initiation. Taken together, these results show that synergid cells possess typical characteristics of glandular cells for protein secretion while neighboring egg cells are well prepared for embryonic initiation.

Remarkably, 28 of the top 30 synergid cell genes (93.3%) show synergid-specific/predominant expression patterns (Supplementary Data 4). This high percentage of cell-specific/predominant expression of the most abundant transcripts was not observed in egg cells or zygotes, supporting the notion of highly specialized functions of flowering plant synergid cells. Moreover, the top synergid cell-specific/predominant genes exhibit extraordinarily high transcript levels, with 21 of them showing FPKM (Fragments Per Kilobase of exon model per Million mapped fragments) values greater than 10,000. We found that among the top 30 synergid cell genes, nearly half (14 out of 30 genes) encode small secreted peptides (Supplementary Data 4), including ZmESs, which functions in pollen tube rupture during pollen tube reception[21,27], and other synergid-specific/predominant cysteine-rich peptides (CRPs) with suggested functions in cell-cell recognition during double fertilization[28]. Notably, one-third (10 out of 30 genes) encode secreted cell wall modifiers, including four pectin methylesterase inhibitors (PMEIs) and three pectate lyases (PLs). We selected two previously undescribed genes (one encoding an unknown CRP and one encoding a PMEI) for validation by expressing *pSC::SC-GFP* constructs in transgenic lines. Being consistent with the cell-specific gene expression data, both of them were strongly and prominently expressed in synergid cells, and were secreted to the micropylar region of the mature ovule (Fig. 2c, d). Further characterization of these abundantly and predominantly expressed secreted proteins will elucidate whether they are involved in facilitating double fertilization and ensuring reproductive success in the monocot and crop plant maize.

### Loss of synergid-specific GRN during synergid degeneration

Analysis of the dynamics of gene expression patterns from the cell- and stage-specific RNA-seq data revealed that the majority of the top 200 synergid-expressed genes were substantially decreased after fertilization (Fig. 2e and Supplementary Data 5). To gain a better understanding of synergid-specific GRNs and their regulation during the entire process of synergid elimination, we investigated the most abundant synergid-expressed transcription factors (TFs). In accordance with the observation that most highly expressed genes in

synergid cells showed synergid-specific/predominant expression patterns (Supplementary Data 4), we found that eight of the top 10 TFs were also specifically or predominantly expressed in synergid cells (Supplementary Data 6). Notably, the top five most strongly expressed TFs exhibited synergid-specific expression patterns (Supplementary Data 6) and were significantly downregulated shortly after fertilization at 12 HAP (Fig. 2f), thus about 6 h earlier than PCD activation, which occurs at around 18 HAP (see below). The decay of mRNAs for two of the TFs was reduced between 12 HAP and 18 HAP and further continued like for the other TFs after 18 HAP (Fig. 2f and Supplementary Data 6). In particular, the expression of *ZmMYB98* was substantially decreased from a 1200 FPKM value in synergid cells to 98 in persistent synergid cells shortly before cell elimination.

Based on the consistent overall expression trend, we next asked whether there is a positive regulation between the top synergid-specific TFs and the top synergid-specific/predominantly expressed genes. As *MYB98* was identified as the most strongly expressed TF gene in maize synergid cells and was previously shown to positively regulate a battery of synergid-expressed genes in *Arabidopsis*[29], we hypothesized that it could regulate other most highly expressed synergid-specific/predominant genes (as well as itself). Therefore, we established a dual-luciferase assay to test whether ZmMYB98 can activate a selected group of the most highly expressed synergid-specific/predominant genes (Fig. 2g). As shown in Fig. 2h, ZmMYB98 stimulated the activities of nearly all selected gene promoters except that of two genes involved in cell wall modification. All secreted peptide genes, including the most strongly expressed CRP genes with potential roles in pollen tube attraction[12,30] and *ZmES4*[21] were activated by ZmMYB98. Moreover, four cognate genes encoding secreted cell wall modifier PMEIs, with proposed function in pollen tube growth arrest and reception[31], exhibited strong induction by ZmMYB98 (Fig. 2h). In addition to secreted peptides and cell wall modifiers, two genes encoding synergid cell-predominant ER protein orthologs (ZmTIN1 and ZmRTNL), a gene encoding a synergid cell-specific plasma membrane blue copper protein (ZmBCP), and a gene encoding a synergid cell-specific cytoplasm annexin (ZmANN4) (Supplementary Data 4), were also positively regulated by ZmMYB98. Since the *ZmMYB98* promoter region itself contains several putative MYB98 binding sites (TAAC)[29], we then tested whether ZmMYB98 could regulate its own expression. The transactivation assay showed that *ZmMYB98* is indeed autoregulated (Fig. 2h and Supplementary Fig. 4).

To further test this regulation, we investigated the expression and subcellular localization of ZmMYB98-activated ZmSC-GFP fusion proteins under the control of their endogenous promoters in a heterologous dicot system *in planta* (Supplementary Fig. 4). In the absence of ZmMYB98, GFP signals could not be detected in transfected tobacco epidermal cells. In contrast, ZmMYB98 co-expression efficiently promoted the activity of synergid-specific/predominant promoters for ZmSC-GFP expression and secretion as indicated by GFP signals localized at the apoplastic space (Supplementary Fig. 4). These results confirm that the most highly expressed synergid-specific/predominant genes require ZmMYB98 for expression, and ectopic expression of

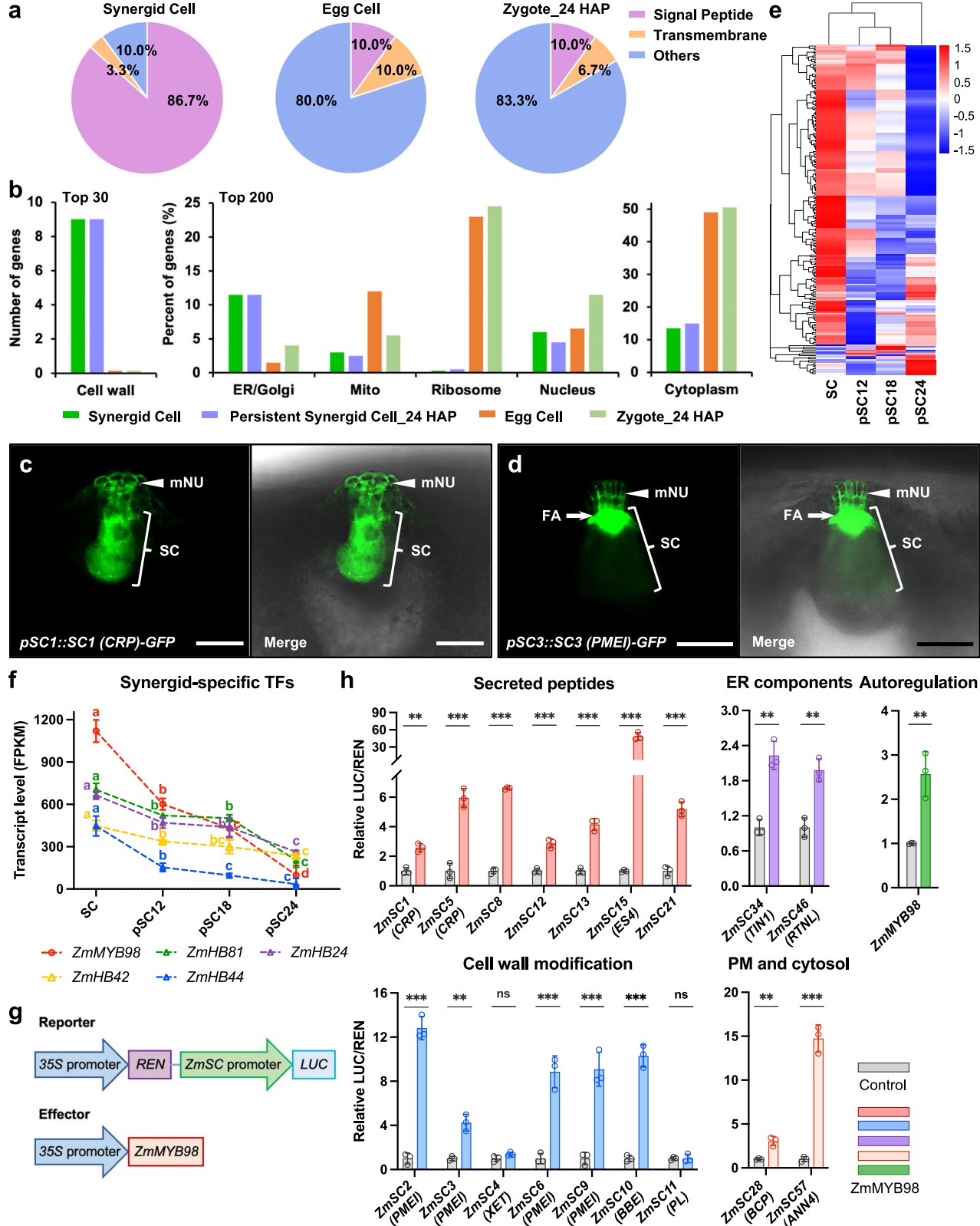

ZmMYB98 in tobacco epidermal cells is sufficient to activate the expression of maize synergid-specific genes in these cells. We therefore conclude that ZmMYB98 controls the expression of most abundantly and predominantly expressed genes in synergid cells. This synergid-specific GRN is significantly downregulated immediately after fertilization (i.e., more than 6 h before the activation of persistent synergid PCD) and nearly shut off shortly before cell elimination.

## ZmRALF12 is required for ROS production at the filiform apparatus adjacent region

ROS are signaling molecules for many basic biological processes, and moreover, they act as main triggers for physiological or programmed pathways for cell death during plant development, reproduction, and stress responses[32–34]. To determine whether ROS are involved in the degeneration process of the persisting synergid cell, we stained maize

**Fig. 2 | Synergids are glandular cells generating many peptides via a ZmMYB98 controlled GRN. a** Proportion comparison of the 30 most highly expressed genes in synergid cells before pollination. 86.7% genes encode proteins containing predicted signal sequences for targeting to the secretory pathway. Egg cells and zygotes at 24 HAP are included for comparison. **b** Subcellular distribution of most abundant gene products in indicated cells. Localization of the fusion proteins SC1-GFP (a CRP) (**c**) and SC3-GFP (a PMEI) (**d**) in synergid cells and cell walls of micropylar nucellus cells in mature embryo sacs before pollination. Arrowheads point towards extracellular signals at the micropylar nucellus region. **e** Heatmap showing expression pattern of the TOP200 synergid cell-expressed genes during progression of synergid degeneration. **f** Expression analysis of the most abundant synergid cell-expressed transcription factors during synergid degeneration. For statistical analysis, count data at the gene level were analyzed with DESeq2[64]. Stage-to-stage comparisons were performed and corrected for multiple testing over all genes and cell stage comparisons using false discovery rate (FDR). Letters indicate significant differences between developmental stages (adjusted $P < 0.05$). **g** Scheme of reporter and effector vectors for dual-luciferase transactivation assays presented in (**h**). **h** Transactivation assay showing that ZmMYB98 positively regulates a comprehensive set of most highly expressed synergid-specific/predominant genes. Data from previous publications[22,69,70] were used to identify synergid-specific/predominant genes (Supplementary Data 4 and 6). Relative ratio of firefly luciferase (LUC) to *Renilla* luciferase (REN) was used to determine relative promoter activation activity. Results are an average of three independent experiments. Statistical significance is determined by two-tailed Student's *t*-test (**$P < 0.01$, ***$P < 0.001$) and the error bars show standard deviation (SD). Source data and further statistical analysis are provided in the source data file. Observations of GFP signals in *pZmSC1::ZmSC1-GFP* (**c**) and *pZmSC3::ZmSC3-GFP* (**d**) were performed three times with similar results. FA filiform apparatus, mNU micropylar nucellus, SC synergid cell. Scale bars, 50 μm.

ovule tissues at different time points before and after fertilization, with the general ROS dye 2′,7′-dichlorofluorescein diacetate (DCFH-DA). Before pollination, ROS accumulation was detected in epidermal micropylar nucellus cells and was almost absent from mature synergid cells (Fig. 3a). However, at 20 HAP (-12 HAF), we detected very intense DCFH-DC staining in the persisting synergid cell, with a maximum ROS accumulation adjacent to the plasma membrane (Fig. 3b–d). To gain more detailed information about ROS production, microdissected whole embryo sacs containing a visible filiform apparatus at the micropylar end were double-stained with a mitochondria-specific fluorescence dye (Mito-Tracker Red) and the ROS marker DCFH-DA. We observed that before pollination, synergid mitochondria constitutively accumulate alongside a compartment that we name filiform apparatus adjacent region (FAAR). ROS was not detectable in the FAAR (Fig. 3e). At 20 HAP, strong granular ROS signals were present, prominently at this polarly localized mitochondria-enriched FAAR (Fig. 3f). These findings indicate that fertilization induces granular ROS production at the polar mitochondria-enriched region, and it accumulates to high levels before persistent synergid degeneration.

Plant NADPH oxidases, also known as respiratory burst oxidase homologs (RBOHs), have been called 'the engines of ROS signaling'[34]. Two *RBOH* genes, *ZmRBOH3* and *ZmRBOH4*, were expressed in maize synergid cells and were further transcriptionally activated at 12 HAP (Fig. 3g), indicating that major components of the ROS production machinery were present. RBOHs were previously shown to be activated by Rapid Alkalinization Factor (RALF) signaling peptides involving also members of the *Catharanthus roseus* RLK1-like (CrRLK1L) receptor kinase family and their GPI-anchored LORELEI-like (LLG) co-receptors[35,36]. Among the 24 RALF genes in maize[25], only *ZmRALF12* was highly induced during persistent synergid degeneration (Fig. 3h and Supplementary Data 7). Additionally, two CrRLK1L and one LLG encoding genes (*ZmFERL1*, *ZmFERL16* and *ZmLLG3*) that potentially form a heterotypic receptor complex with the mobile RALF ligand[25] were activated shortly after fertilization and continuously increased (with the exception of *ZmFERL1*) indicating that a functional RALF-CrRLK1L-LLG complex can be formed before synergid cell degeneration. This is supported by yeast two-hybrid assays showing that ZmRALF12 interacts with these co-expressed and upregulated receptors (Supplementary Fig. 5). We further performed immunological analysis using antibodies against ZmRALF12. Consistent with the cell- and stage-specific RNA-seq data and the predicted secreted subcellular location, ZmRALF12 peptides were undetectable before fertilization and detected at the filiform apparatus only from 18 HAP, and with a significant increase at 24 HAP at both the filiform apparatus and the FAAR (Fig. 3i–l). We therefore hypothesized that the fertilization-induced ZmRALF12 peptide ligand is the limiting factor that triggers ROS production and subsequent oxidative burst in the persisting synergid cell. To test this hypothesis, we generated *zmralf12* mutants using the CRISPR-Cas9-mediated genome editing system. We obtained the loss-of-function mutant *zmralf12⁻/⁻*

(Supplementary Fig. 6) and investigated ROS production after fertilization. In contrast to wild-type plants (Fig. 3f), ROS were not induced at the mitochondria-accumulating FAAR at 20 HAP in *zmralf12* mutants (Fig. 3m). Therefore, we conclude that fertilization-induced ROS production, prominently at the mitochondria-enriched FAAR, is correlated with ZmRALF12 production and accumulation in this area.

## ZmRALF12 triggers mitochondrial ROS production

We next investigated whether ZmRALF12 indeed has the ability to induce ROS production in vivo in synergid cells. Using an optimized ovule cultivation system, we applied each 10 μM ZmRALF12 and two pollen tube-expressed ZmRALF1 and ZmRALF2 peptides[25] for 30 min to the ovule culture medium containing unpollinated ovule tissues. ROS accumulation and mitochondria localization were detected using DCFH-DC and Mito-Tracker Red double staining. In comparison to ZmRALF1 and ZmRALF2, ZmRALF12 (gene ID Zm00001eb410350) treated ovule tissues displayed strong ROS accumulation at the FAAR (Fig. 4a and Supplementary Fig. 7a–c), comparable to signals in persistent synergid cells from pollinated ovule tissues at 20 HAP (Fig. 3f).

In order to attain a higher resolution to discern subcellular localization, we then analyzed mitochondria localization and ZmRALF12 response in microdissected synergid cells. The *pZmSC3::ZmSC3-GFP* line was used in combination with Mito-Tracker Red staining to trace the filiform apparatus and the FAAR during manual synergid cell isolation. Consistent with observations using the B73 wild type (WT) inbred line (Fig. 3e), mitochondria of *pZmSC3::ZmSC3-GFP* plants were accumulated at the FAAR in synergid cells (Supplementary Fig. 7d). This mitochondria-enriched region is maintained in synergid protoplasts after manual isolation (Supplementary Fig. 7e–g), which exhibits efficient and specific ZmRALF12 response shortly after peptide ligand application (Supplementary Fig. 7h). These findings confirm that manual cell dissection does not affect synergid functionality in polar mitochondria distribution and ZmRALF12 response.

We thus conducted ZmRALF12 treatment (4 μM for 30 min) with manually isolated synergid cells before pollination and double-stained the cells with Mito-Tracker Red and DCFH-DA. We found both, ROS and mitochondrial signals, showing prominent localization as accumulated particles at the FAAR (Fig. 4b). This cellular synergid domain was previously reported to contain the FER-LLG receptor complex in the model plant *Arabidopsis*[23,24]. Colocalization of ROS production sites and mitochondria was revealed by the overlap of signals resulting in yellow staining and a strong correlation of relative fluorescence intensity plots (Fig. 4c). The same result was obtained when double staining was performed on microdissected persistent synergid cells at 20 HAP (Fig. 4d, e). Notably, in both cases, the strongest ROS production area was at the FAAR, which is in close proximity to the cellular ZmRALF12-receptor interacting region. A decrease in subcellular ROS signal intensity correlates well with the distance from the micropylar FAAR (Fig. 4c, e). Collectively, we conclude that fertilization-induced

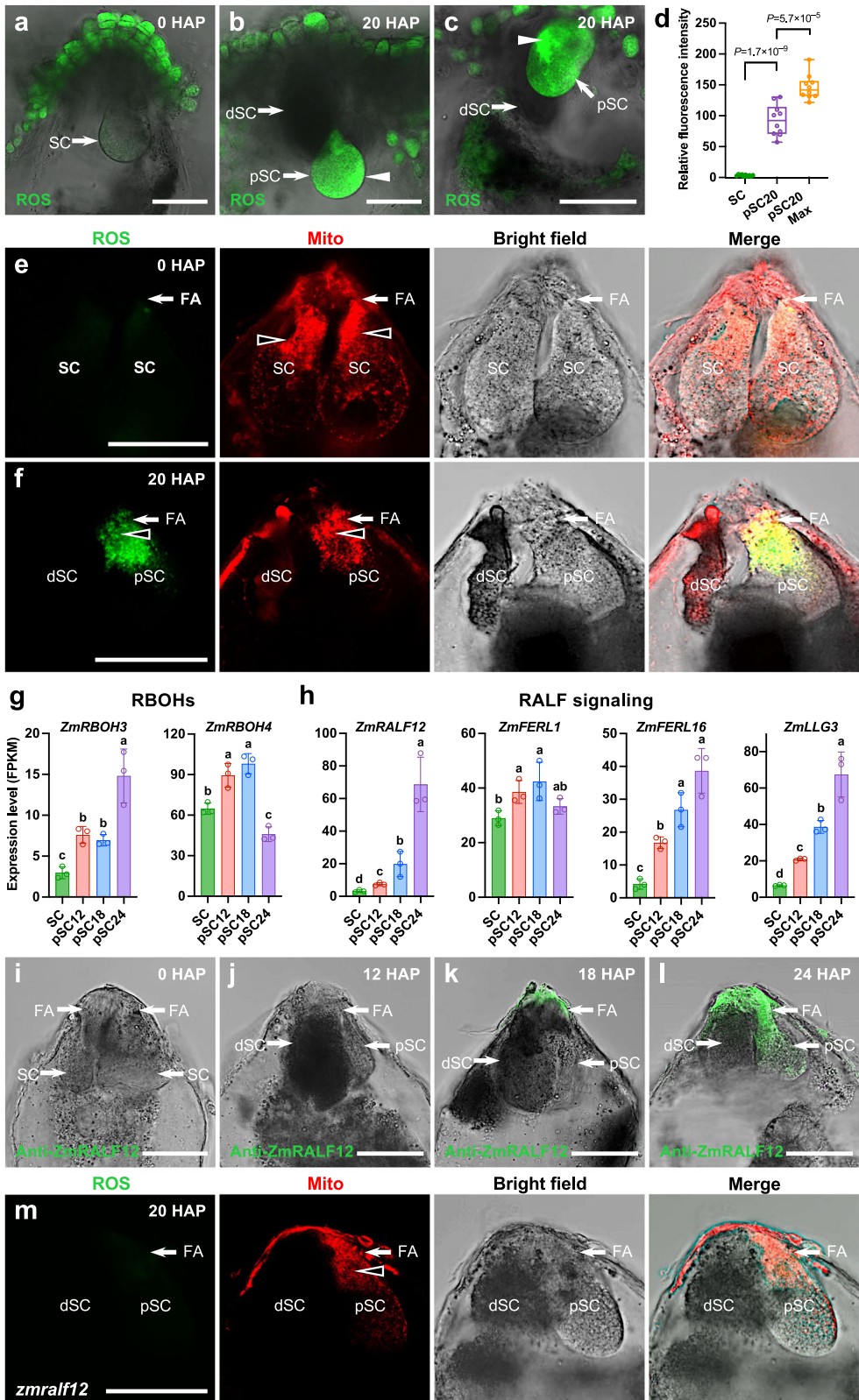

ZmRALF12 triggers ROS production and accumulation in synergid mitochondria from the filiform domain of synergid cells.

To further decipher the underlying process that leads to mitochondrial ROS production, we explored which ZmRALF12 downstream effectors might regulate the mitochondrial ROS response. Since two NADPH oxidases (RBOH3 and RBOH4)−potential signaling pathway effectors[10,36]−were upregulated in persistent synergid cells after

fertilization (Fig. 3g), we then determined whether RBOHs, with predicted localization at the plasma membrane, are involved in mitochondrial ROS production. Preincubation with VAS2870, a specific inhibitor of NADPH oxidases[37], abolished ROS production in both, mitochondria and cytosol (Supplementary Fig. 8a, b). This strongly supports the key role of RBOHs in promoting mitochondrial ROS production, being consistent with a recent study showing that the

**Fig. 3 | ROS production in the persisting synergid cell is correlated with fertilization-induced accumulation of ZmRALF12 at the filiform apparatus.**
**a** DCFH-DA staining showing ROS accumulation in epidermal micropylar nucellus cells and its absence inside the mature embryo sac. **b, c** At 20 HAP strong staining is detectable inside the persistent synergid cell. White arrowheads point to ROS maxima in persistent synergid cells. **d** Quantification of ROS fluorescence intensity in synergid cells at stages shown in (**a**–**c**) ($n = 10$). **e, f** DCFH-DA and Mito-Tracker Red double-staining showing mitochondria constitutively accumulating alongside the filiform apparatus (FA) adjacent region (FAAR) in synergid cells (black arrowheads). ROS are undetectable in synergid cells before pollination (**e**) and accumulate at the mitochondria-enriched FAAR in the persistent synergid cell at 20 HAP (**f**). **g, h** Expression of RBOH and RALF peptide/receptor genes during fertilization and persistent synergid degeneration. Transcript levels are shown as FPKM values (means ± SD) of three biological replicates. For statistical analysis, count data at the gene level were analyzed with DESeq2[64]. Stage-to-stage comparisons were performed and corrected for multiple testing over all genes and cell stage comparisons using false discovery rate (FDR). Letters above bars indicate significant differences (adjusted $P < 0.05$). **i–l** Immunostaining showing secreted ZmRALF12 peptides at the FA from 18 HAP (**k**) and increasing amounts at 24 HAP at both, FA and FAAR regions. **m** DCFH-DA and Mito-Tracker Red double-staining in *zmralf12* mutant embryo sacs. ROS are not induced at the mitochondria-accumulating FAAR in the persistent synergid cell at 20 HAP. In box-and-whisker plots, center lines represent the 50th percentile; bottom and top of each box indicate the 25th and 75th percentiles, respectively; whiskers represent minimum and maximum, respectively. Statistical significance is determined by two-tailed Student's *t*-test. Source data and adjusted *P* values (in **g, h**) are provided in the source data file. DCFH-DA and Mito-Tracker Red double-staining of intact synergid cells in embryo sacs 0 HAP (**e**) and 20 HAP (**f**), as well as ZmRALF12 immunostaining (**i–l**), and DCFH-DA and Mito-Tracker Red double-staining of *zmralf12* mutant embryo sacs 20 HAP (**m**) were repeated three times with similar results. dSC degenerated receptive synergid cell, FA filiform apparatus, pSC persistent synergid cell, SC synergid cell. Scale bars: 50 μm.

RBOH machinery acts in $H_2O_2$ accumulation in guard cell mitochondria to facilitate stomatal closure[37]. Previous studies in animal systems have shown that ROS production and signaling are intimately associated with $Ca^{2+}$ channel activation, enabling $Ca^{2+}$ flux into the cytosol or a particular subcellular compartment (e.g., mitochondria). Elevated $Ca^{2+}$ uptake in mitochondria is intimately linked to respiratory chain activity stimulation, leading to higher amounts of mitochondrial ROS production[38]. In support of this scenario, we found that $H_2O_2$, the product of RBOHs and the major ROS in unicellular and multicellular organisms[39], can effectively induce mitochondrial ROS accumulation in synergid cells. Furthermore, both ZmRALF12-triggered RBOH-dependent and $H_2O_2$-induced mitochondrial ROS accumulation is $Ca^{2+}$ channel-dependent (Supplementary Fig. 8c–f). Since ABA signaling is also upregulated during persistent synergid degeneration (see below), we further tested whether ABA could induce ROS production in synergid mitochondria. In contrast to ZmRALF12, ABA treatment did not trigger mitochondrial ROS production in synergids (Supplementary Fig. 8g, h). Taken together, we conclude that ZmRALF12 specifically triggers mitochondrial ROS production at the FAAR, and this process is dependent on RBOH and $Ca^{2+}$ channel activity.

### ZmRALF12-triggered mitoROS production mediates oxidative stress and dPCD in the persisting synergid cell

To further explore the temporal and spatial dynamics of ZmRALF12-induced ROS production, we generated a time series of ZmRALF12 treatment on synergid cells, ranging from 0 to 40 min. Remarkably, already at 5 and 10 min, respectively, ZmRALF12 triggered granular ROS accumulation at the mitochondria-enriched FAAR (Supplementary Fig. 9a), demonstrating rapid RALF-induced mitochondrial ROS production. Extended periods of ZmRALF12 incubation led to significantly elevated granular ROS accumulation at this region and an ultimate granular ROS expansion throughout the whole synergid cell (Supplementary Fig. 9a, b). Similarly, increased ZmRALF12 concentrations correlated with higher ROS levels at the FAAR and the same trend of granular ROS diffusion towards their opposite cellular domain (Supplementary Fig. 9c, d). These results show that fertilization-induced synergid cell-derived autocrine signal ZmRALF12 rapidly (within minutes) triggers mitochondrial ROS accumulation at the FAAR, and mediates high ROS levels in a time- and concentration-dependent manner.

Consistent with a strong increase of *ZmRALF12* expression in synergid cells from 18 to 24 HAP and the observed mitochondrial ROS accumulation at 20 HAP, a subset of genes involved in peroxisomal functions including NAD+ import (*ZmPXN*), 2-hydroxy acid oxidation (*ZmGLO3*), very long-chain fatty acids (VLCFAs) synthesis for peroxisomal metabolization (*ZmKCR1*) and fatty acid oxidation (*ZmSDP1, ZmECH2,* and *ZmKAT5*), were strongly activated in the persisting synergid cell after 18 HAP (Supplementary Fig. 10). These findings

further indicate that the onset of oxidative burst occurs with some delay after fertilization, which ultimately provokes a shift from the compartmental redox balance towards oxidative stress. Previous studies in animal systems demonstrated that the switch from redox balance to oxidative stress is largely mediated through $Ca^{2+}$ signaling[40]. In accordance with an oxidative stress response status in the persistent synergid cell at 24 HAP (Fig. 4f–i), we identified significant upregulation of genes encoding plasma membrane and ER-membrane $Ca^{2+}$ channels (*ZmCSC1* and *ZmTMCO1*), mitochondrial inner membrane $Ca^{2+}$ uniporter that mediates $Ca^{2+}$ uptake into mitochondria (*ZmMCU2*), and $Ca^{2+}$ signaling components involved in senescence regulation (Supplementary Fig. 11). Since ZmRALF12-induced mitochondrial ROS accumulation is $Ca^{2+}$ channel-dependent (Supplementary Fig. 8), the strong upregulation of these $Ca^{2+}$ channel genes would further accelerate mitochondrial ROS production in the persisting synergid. The concomitant induction of gene batteries for oxidative stress response and senescence (Fig. 4f, g) further confirms the shift from a balanced to an oxidative stress status in persisting synergid cells at ~24 HAP.

In contrast to the upregulation of oxidative stress response ROS-scavenging enzymes that are targeted to the extracellular matrix, mitochondrial antioxidant enzymes, which protect cells from apoptosis[41,42], are specifically downregulated after fertilization and during the progressing of synergid degeneration (Fig. 4f). By using monodansylcadaverine (MDC) staining, we further observed active autophagic processes and vacuole accumulation shortly (i.e., only ~1–2 h) before persistent synergid elimination (Fig. 4h, i). Notably, autophagosomes are also most abundant in the FAAR, where ZmRALF12-triggered mitochondrial ROS production and oxidative stress/damage are most evident (Fig. 4i). This is consistent with the activation of a series of mitophagy-related genes known to be involved in autophagic removal of oxidatively damaged mitochondria[43] (Fig. 4g and Supplementary Fig. 12a). Furthermore, ZmRALF12, but not ZmRALF2, can effectively trigger autophagosome production and vacuole accumulation in synergid cells (Fig. 4j, k), comparable to persistent synergid cells at 24 HAP (Fig. 4i). Collectively, these results show that the peptide ligand ZmRALF12 triggers mitochondrial ROS production after successful fertilization accompanied with downregulation of mitochondrial antioxidant enzymes. This mediates the switch to oxidative stress and senescence thereby activating the dPCD pathway in the persisting synergid cell.

### ZmRALF12-triggered mitoROS and ethylene promote sensitized synergid cell death

Since mitochondrial antioxidant enzymes are specifically downregulated after fertilization (Fig. 4f), persistent synergid cells appear to be less protected and more sensitive to mitochondrial ROS accumulation. We therefore investigated whether the absence of

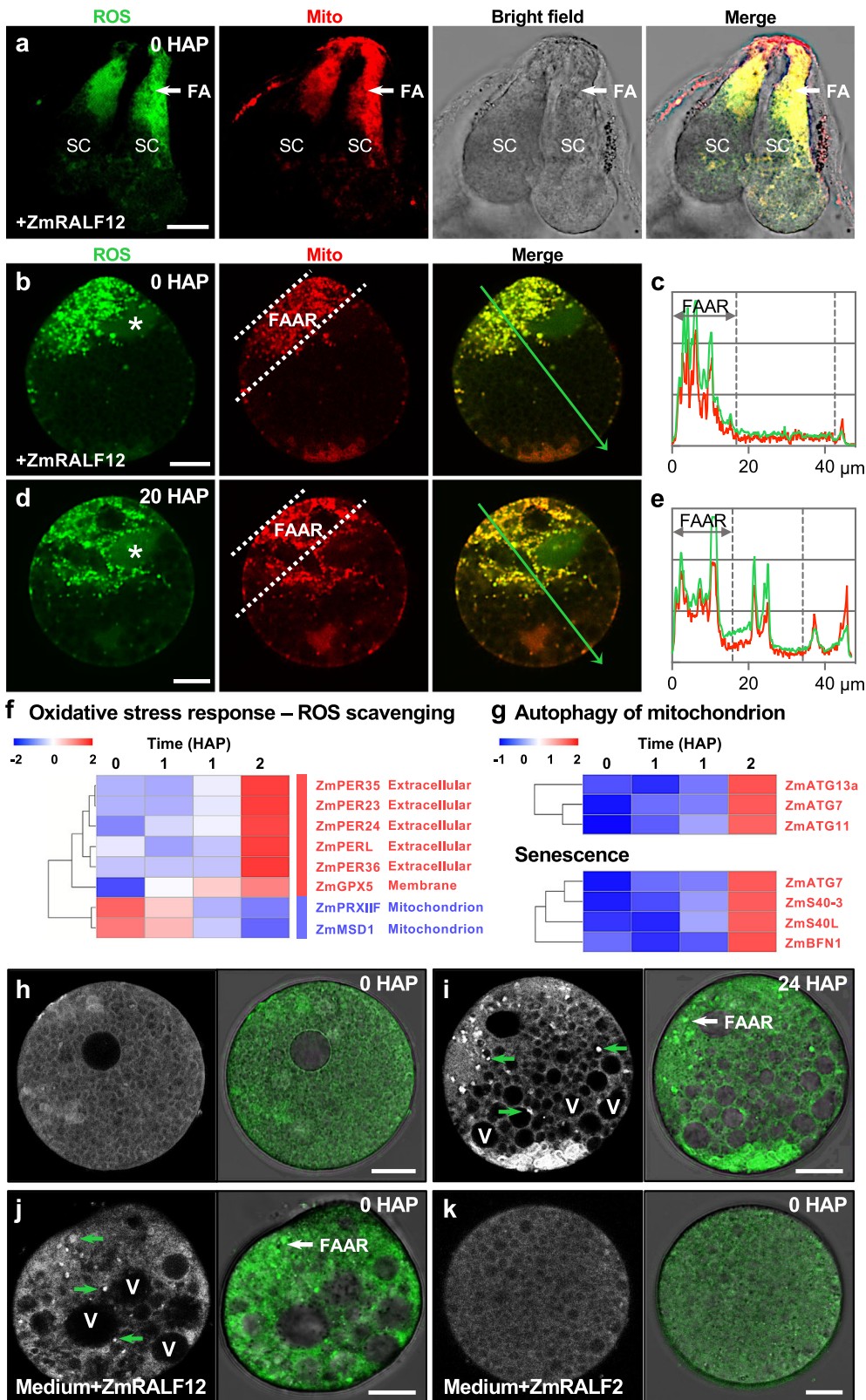

mitochondrial ROS production in *zmralf12* mutants indeed affects persistent synergid degeneration. Due to the expression of *ZmRALF12* in vegetative tissues[25], *zmralf12* mutants showed a dwarf phenotype resulting limited number of ovaries and kernels (Supplementary Fig. 6c–e). Therefore, statistical analysis of persisting synergid degeneration phenotype with *zmralf12* mutants was not applicable. We thus used an alternative approach by manipulating the persistent

synergid environment to mimic the ROS deficiency phenotype of *zmralf12* mutants. Ovules were harvested at 16 HAP and cultivated in an optimized culture medium or medium supplied with ROS scavenger CuCl₂ (5 mM). Compared to the control, post-fertilization ROS scavenger application (mimicking a *zmralf12* mutant) significantly delayed persistent synergid cell death/elimination (Fig. 5a–c). Taken together, ZmRALF12-triggered mitochondrial ROS production in the

**Fig. 4 | Fertilization-induced autocrine ZmRALF12 signal mediates mitochondrial oxidative stress and synergid apoptosis. a** ZmRALF12 application assay using an optimized ovule cultivation system. Sectioned ovules are double-stained with DCFH-DA and Mito-Tracker Red to visualize the ZmRALF12 response and mitochondria localization. ZmRALF1 and ZmRALF2 application were negative (Supplementary Fig. 7). ZmRALF12 application (4 μM for 30 min) triggers high levels of mitochondrial ROS accumulation at the FAAR in manually isolated synergid cells (**b**, **c**), comparable to persistent synergid cells isolated at 20 HAP (**d**, **e**). Asterisks indicate moderate ROS accumulation in synergid nuclei. Merged images of ROS accumulation and a mito-tracker are as indicated (**b**, **d**). Intensity plots (**c**, **e**) were generated along the green arrows in (**b**, **d**). **f** In contrast to the activation of extracellular targeted ROS scavenging enzymes, mitochondrial ROS scavenging enzymes are downregulated after fertilization and continually decrease during synergid degeneration. **g** Mitophagy and senescence-related genes are strongly activated at 24 HAP. **h**, **i** Monodansylcadaverine (MDC) staining showing an intact synergid cell before fertilization (**h**) and active autophagic processes and vacuole accumulation in the persistent synergid cell before its elimination (**i**). **j**, **k** ZmRALF12, but not ZmRALF2, can effectively trigger autophagosome production and vacuole accumulation in synergid cells, comparable to persistent synergid cells at 24 HAP (**i**). Green arrows point to autophagosomes transported to vacuoles. The white arrow shows autophagy maxima at the mitochondrial ROS accumulating FAAR. ZmRALF12 application assays (**a**, **b**), DCFH-DA and Mito-Tracker Red double staining of persistent synergid cells at 20 HAP (**d**), MDC staining of synergid cells at 0 HAP (**h**) and persistent synergid cells at 24 HAP (**i**), as well as MDC staining after ZmRALF12 (**j**) or ZmRALF2 (**k**) treatment were repeated three times with similar results. FA filiform apparatus, FAAR filiform apparatus adjacent region, SC synergid cell. Scale bars, 20 μm (**a**) and 10 μm (**b**, **d**, **h**–**k**), respectively.

persisting synergid cell crucially determines timely persistent synergid cell death and elimination.

ROS signaling is interconnected with the response to several phytohormones, especially the gaseous hormone ethylene. Production of ethylene and the activation of ethylene responses have been reported to belong in vegetative development as the first events following the accumulation of ROS[44,45]. In *Arabidopsis*, it has been previously shown that ethylene signaling is required for the degeneration of the persistent synergid cell and the establishment of a pollen tube block[18,19,46]. To dissect the contributions of ethylene and ethylene signaling to synergid cell death, we therefore first investigated the dynamic expression changes of genes related to ethylene biosynthesis and signaling. The ethylene biosynthetic pathway consists of two well-defined enzymatic steps: first, ACC synthase (ACS) catalyzes the conversion of *S*-adenosyl methionine (SAM) to the intermediate 1-aminocyclopropane-1-carboxylic acid (ACC), and second, ACC oxidase (ACO) oxidizes ACC to generate ethylene[47,48]. Our cell- and stage-specific RNA-seq data set revealed that both *ACS* and *ACO* genes were expressed throughout the processes of fertilization and persistent synergid degeneration, but in contrast to *ACSs*, *ACOs* (*ZmACO5* and *ZmACO20*) showed a significant increase at 24 HAP (Fig. 5d and Supplementary Data 7). Accordingly, the key regulators of ethylene signaling, EIN3 and EIL1, which are sufficient for the activation of the ethylene-response pathway[49], were de novo expressed after fertilization and showed an enormous increase at this later stage (Fig. 5d). In consequence, eight potential downstream AP2/ERF transcription factors were specifically activated in a similar pattern (Fig. 5d). Notably, in neighboring zygotes at 24 HAP, all eight AP2/ERF transcription factor genes as well as those encoding ACOs remained not expressed or unactivated[22] indicating that the synergid cell-specific GRN is inactive in the neighboring zygote.

Given the specific activation of ethylene biosynthesis, signaling, and responses in the persisting synergid cell (Fig. 5d), and the fact that the neighboring zygote and endosperm are seemingly unaffected by gaseous ethylene, we hypothesized that increased ethylene production may trigger cell death exclusively in sensitized synergid cells. In *Arabidopsis*, microinjection of the ethylene precursor ACC into the female gametophyte led to premature synergid disintegration indicating that synergid cells appear to be more sensitive to ACC treatment and ethylene signaling[18]. However, whether ethylene itself plays a key role in synergid degeneration or components of the downstream signaling pathway is controversial, as ethylene production is not necessarily required for the death of synergid cells[19]. To determine the effect of post-fertilization ethylene production on female gametophytic cells, we therefore performed ethylene application experiments with mature virgin maize ovules to mimic an ethylene burst environment after fertilization. In the mock-treated control, intact synergid nuclei appeared large and round, and the synergid cell status was comparable to untreated materials (*n* = 119; Fig. 5e). By contrast, in about 40% of ethylene-treated ovules (*n* = 163), synergid cells exhibited deformed or condensed nuclei, or showed a highly degenerated status as evidenced by strong autofluorescence (Fig. 5f–h). Strikingly, the surrounding female gametophytic cells (i.e., egg cell and central cell) were intact and displayed normal nuclei morphologies (Fig. 5f, g). These results imply that synergid cells are especially primed for ethylene-triggered cell death and that the ethylene signaling pathway is highly activated in the persisting synergid cell a few hours before its death. In conclusion, together with elevated ZmRALF12-triggered mitochondrial ROS accumulation and oxidative stress-mediated dPCD, these two pathways act synergistically to efficiently promote sensitized synergid cell death after fertilization.

## PCD of the persistent synergid cell is activated with delay after fertilization

During manual isolation of synergid cells at defined stages during PCD, we observed abrupt removal of the persistent synergid cell after about 25 HAP (Fig. 1l and Supplementary Fig. 1c, d). To investigate the cytological characteristics of this event, we analyzed ovules from maize inbred line B73, as well as two synergid marker lines *pZmES4::ZmES4-GFP* and *pZmSC1::ZmSC1-GFP*, respectively. Before pollination, the synergid cells at the micropylar end of the female gametophyte exhibited clear cell boundaries, with a large and round nucleus situated in the micropylar region of the cell (Fig. 6a). At 24 HAP, the persistent synergid cell still appeared intact and its nucleus displayed largely normal morphology (Fig. 6b). However, after 25 HAP, we observed apparent disappearance of the cell boundary between the persistent synergid cell and the endosperm. Besides, a condensed or disorganized nucleus was located at the micropylar end of the embryo sac, and was eliminated at a slightly later stage (Fig. 6c–e). Detailed analysis of the *pZmSC1::ZmSC1-GFP* marker line further showed that at approximately 25-26 HAP, GFP signals diffused from the persistent synergid cell towards the endosperm, demonstrating the fusion between the persistent synergid cell and the primary endosperm (Fig. 6f, g). This led to a rapid decrease of GFP signals and indicated the removal of the persistent synergid cell (Fig. 6f, g). The resulting cytoplasmic continuity between the persistent synergid cell and the endosperm was further supported by observation of manually isolated embryo sacs at 28 HAP. After incubation in 12% mannitol solution for several minutes, they showed moderate plasmolysis of a syncytium at the micropylar region of the embryo sac (Supplementary Fig. 12). These observations show that—like in *Arabidopsis*[15]—the phenomenon of synergid-endosperm fusion (SE fusion) occurs also in the evolutionarily distant cereal crop maize, to quickly remove the persistent synergid cell and thus to terminate pollen tube attraction and other functions of synergid cells.

Notably, elimination of the persistent synergid has been discussed already before to occur via dPCD[6,7,12,15,18,19]. However, the molecular regulation that ensures the timely activation and execution of PCD processes remained largely elusive. Therefore, we next investigated the expression of maize PCD machinery components

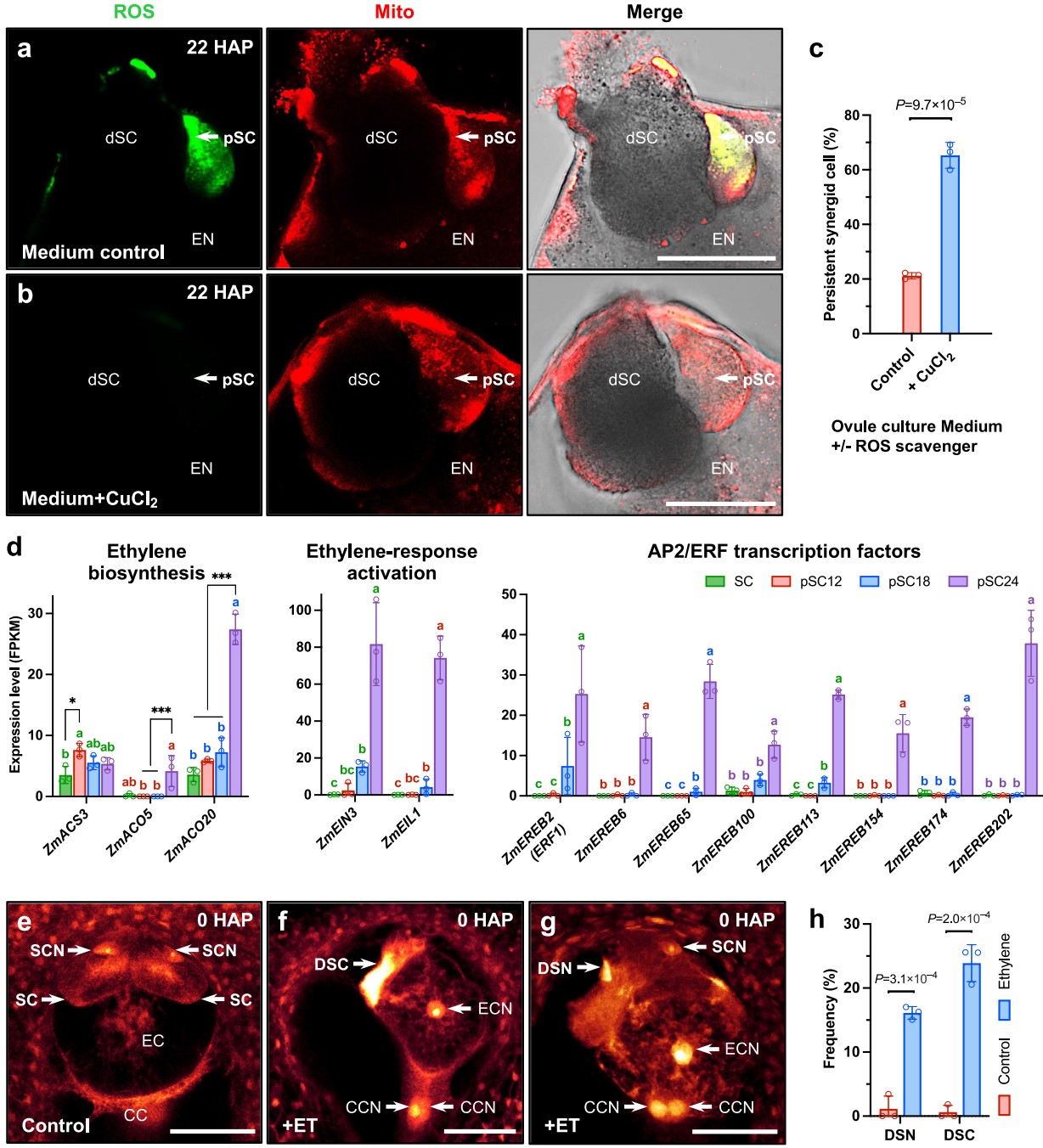

**Fig. 5 | Fertilization-induced ROS and ethylene production act synergistically to promote synergid cell death in maize. a–c** Post-fertilization ROS scavenger application delays persistent synergid cell death/elimination. Ovules were harvested at 16 HAP and cultivated in an optimized culture medium (**a**; control) and medium supplied with the ROS scavenger CuCl$_2$ (5 mM, **b**), respectively. In contrast to control (**a**), ROS was almost non-detectable in persistent synergid cells after CuCl$_2$ application 22 HAP (**b**). **c** Frequency of persistent synergid cell-containing embryo sacs analyzed at 28 HAP (control: $n$ = 28; +CuCl$_2$: $n$ = 30). **d–h** Post-fertilization ethylene production activates the ethylene signaling pathway and promotes synergid cell death in maize. **d** Strong activation of ethylene (ET) biosynthesis and ET-responsive genes at 24 HAP in the persistent synergid cell. Transcript levels are shown as FPKM values (means ± SD) of three biological replicates. For statistical analysis, count data at the gene level were analyzed with DESeq2[64]. Stage-to-stage comparisons were performed and corrected for multiple testing over all genes and cell stage comparisons using false discovery rate

(FDR). Letters above bars indicate significant differences (adjusted $P < 0.05$). For ET biosynthesis genes, significant differential expressions (DEs) are shown (*$\log_2$FC > 1 and adjusted $P < 0.05$, ***$\log_2$FC > 1 and adjusted $P < 0.001$). ET application specifically triggers only synergid cell death in the mature female gametophyte. CLSM of a female gametophyte without ET treatment (**e**; control) and 30 h after ET treatment exhibiting fully degenerated synergid cells (**f**) and degenerated synergid nuclei (**g**), respectively. **h** Frequencies of synergid defects (control: $n$ = 119; +ET: $n$ = 163). Statistical significance (in **c**, **h**) is determined by two-tailed Student's $t$-test, and data are presented as mean values ± SD. Source data and adjusted $P$ values (in **d**) are provided in the source data file. CC central cell, CCN central cell nucleus, DSC degenerated receptive synergid cell by pollen tube reception, DSC degenerated synergid cell by ET treatment, DSN degenerated synergid nucleus by ET treatment, EC egg cell, ECN egg cell nucleus, EN endosperm, pSC persistent synergid cell, SC synergid cell, SCN synergid cell nucleus. Scale bars, 50 μm.

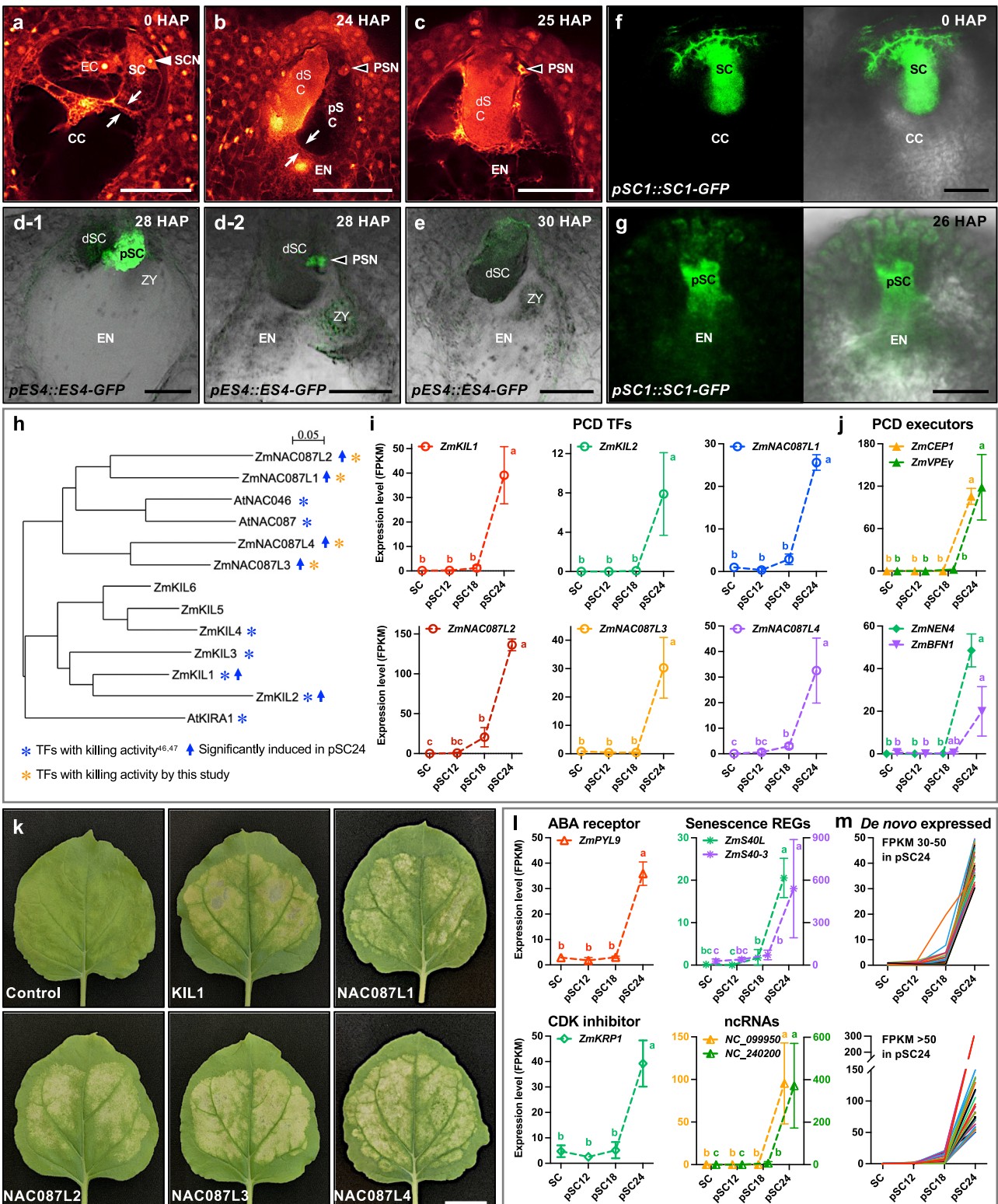

homologous to known *Arabidopsis* and maize PCD transcription factors (TFs) and PCD executors. NAC TFs (<u>N</u>AM: NO APICAL MERISTEM, <u>A</u>TAF1/2: *Arabidopsis thaliana* Activation Factor1 and 2, <u>C</u>UC: CUP-SHAPED COTYLEDON 2) belong to the best-studied TF families in regulating dPCD. ZmKIL1, for example, an ortholog of the *Arabidopsis* NAC TF KIRA1, was recently shown to promote senescence and dPCD in the silk strand base and thus terminate fertility in maize[50]. We found that both, *ZmKIL1* and *ZmKIL2* were de novo activated after 18 HAP (Fig. 6h, i and Supplementary Data 7). ANAC087 and ANAC046 have

been reported to control dPCD in *Arabidopsis* columella root cap cells[51]. Their homologs in maize (*ZmNAC087L1-4*) exhibited a similar delayed and de novo activation pattern in the persisting synergid cell like *ZmKIL1* and *ZmKIL2* (Fig. 6h, i and Supplementary Data 7). We next studied if the expression of these maize homologs can effectively induce cell death. The leaf transfection assay showed that each of these recently identified candidate dPCD-related NAC TFs (i.e., ZmNAC087L1-4) was sufficient to induce senescence and ectopic cell death (Fig. 6i, k) and therefore represents a previously

**Fig. 6 | Programmed cell death (PCD) of the persistent synergid cell is activated after successful double fertilization.** CLSM images of embryo sacs before pollination (**a**) and at 24 HAP (**b**) and 25 HAP (**c**), respectively. White arrowhead shows synergid cell nucleus. Black arrowheads point towards degenerating persistent synergid nuclei. White arrows indicate cell boundaries. **d–g** Fluorescence microscopy to monitor the process of persistent synergid elimination that occurs at about 26–30 HAP. Embryo sacs at indicated time points expressing ES4-GFP and SC1-GFP driven by their endogenous promoters, respectively. The diffusion of GFP signals from persisting synergid cells to the endosperm was observed at 26 HAP (**g**). **h** Phylogenetic tree of NAC family PCD transcription factors from *Arabidopsis* and their maize homologs expressed during synergid degeneration. **i, j** Genes encoding a battery of PCD transcription factors with killing activity and their downstream PCD executors are activated/de novo expressed at 18–24 HAP. **k** *Nicotiana benthamiana* leaves infiltrated with NAC family PCD transcription factors that are significantly induced in persisting synergid cells at 24 HAP.

**l** Activation patterns of selected maize genes encoding homologs of *Arabidopsis* ABA receptor, CDK inhibitor, and senescence regulators involved in cell death regulation. The two most strongly induced ncRNAs with similar activation patterns are shown. **m** De novo expressed unknown genes in the persistent synergid cell showing a PCD-related expression pattern. Data (in **i, j, l**) are presented as mean values ± SD. For statistical analysis, count data at the gene level were analyzed with DESeq2[64]. Stage-to-stage comparisons were performed and corrected for multiple testing over all genes and cell stage comparisons using false discovery rate (FDR). Letters indicate significant differences between developmental stages (adjusted *P* < 0.05). Source data and adjusted *P* values (**i, j, l**) are provided in the source data file. In (**a–g, k**) experiments were repeated three times with similar results and presented are representative images. CC central cell, dSC degenerated receptive synergid cell, EC egg cell, EN endosperm, pSC persistent synergid cell, PSN persistent synergid nucleus, SC synergid cell, SCN synergid cell nucleus, ZY zygote. Scale bars: 50 μm (**a–g**) and 2 cm (**k**), respectively.

uncharacterized key molecular player for driving dPCD in maize and likely other plants.

Upon triggering signals and PCD TF activation, PCD execution and corpse clearance are usually initiated[3]. The cysteine protease CEP1, for example, which is transported to the vacuole and transformed into a mature enzyme before rupture of the vacuole, functions as a key executor in tapetal PCD[52]. Vacuolar processing enzymes (VPEs), major vacuolar proteases with caspase-like activity, can activate other vacuolar hydrolases and thus are in control of tonoplast rupture and vacuole collapse[2,53]. VPEs translocate to the vacuole through the autophagy pathway[54], and autophagy is functionally implicated in corpse clearance during dPCD[55,56]. We searched for maize homologs and found a dramatic de novo activation of *ZmCEP1* and *ZmVPEγ* in persistent synergid cells at 24 HAP (Fig. 6j), being consistent with a clear activation of autophagic processes for PCD execution (Fig. 4h, i). Besides vacuolar proteases, three putative NAC-dependent nucleases (ZmENDO1, ZmNEN1/4) were simultaneously de novo induced during PCD execution and corpse clearance (Fig. 6j and Supplementary Data 7). Taken together, these results show that autophagy and key vacuolar cysteine proteases have been activated for vacuole-related corpse clearance before persistent synergid elimination. Moreover, morphological disorganization of the persistent synergid cell nucleus —another characteristic of PCD—was observed demonstrating that PCD precedes SE-fusion and thus ultimately synergid cell elimination.

Besides the canonical PCD TFs and PCD executors, a subset of genes involved in PCD regulation were also strongly activated. These include orthologs of ABA receptor *PYL9* and CDK inhibitor *KRP1*, senescence regulators, as well as highly induced ncRNAs (Fig. 6l). In total, among the 2727 genes showing significant upregulation after fertilization at 24 HAP (Supplementary Data 8), 379 genes were de novo expressed (only genes with an FPKM ≥ 5 were considered; Supplementary Data 9), including above described PCD regulators, TFs and executors. Notably, nearly all of these genes exhibited a PCD-related expression pattern, that is, not expressed before 18 HAP (-10 HAF) or slightly induced at 18 HAP, but dramatically activated shortly before persistent synergid elimination (Fig. 6m). These include many so far uncharacterized genes, representing very likely emerging components of the molecular machinery responsible for driving dPCD in plants.

## Discussion

During fertilization in flowering plants, two synergid cells undergo cell death in a tightly controlled two-step process, which is essential for reproductive success[5,6,15–19]. Previous studies in *Arabidopsis* have provided significant insights into the mechanisms leading to disintegration of the first (receptive) synergid cell during pollen tube reception[6,10,14,19], while degeneration of the second (persistent) synergid cell remained largely unclear[4,15,19]. By exploring activated signaling pathways before persisting synergid cell elimination, we demonstrated

how fertilization-induced synergid-derived autocrine peptide signal ZmRALF12 specifically triggers mitochondrial ROS production at the FAAR. Accompanied by downregulation of genes encoding mitochondrial antioxidant enzymes, this mediates oxidative stress and activates dPCD in persisting synergid cells before SE fusion. This fusion event, which has so far only been described in *Arabidopsis*[15], is likely conserved in flowering plants. Notably, we show that consistent with the specific and significant activation of *ACOs* in persisting synergid cells, the gaseous plant hormone ethylene itself is capable of inducing death specifically of synergid cells, but not of the surrounding female gametes. This is reflected by the presence of the sensitive ethylene perception, signaling, and response machinery in synergid cells, but not in the surrounding cells. Taken together, we have shown that these two pathways (i.e., ZmRALF12-triggered mitoROS and ethylene production/signaling) work synergistically to ensure the timely disintegration and elimination of persistent synergid cells.

ROS production is generally coupled with a FER-GEF-ROP-RBOH module[10,57]. Therefore, RALF peptides are expected to increase ROS through the activation of plasma membrane-localized RBOHs. There have been no relevant studies revealing RALF-triggered mitochondrial ROS production and its consequences in plants. However, in the present study, we demonstrated that ZmRALF12 can efficiently induce high levels of mitochondrial ROS production prominently at the FAAR in an RBOH- and Ca[2+] channel-dependent manner, which crucially determines timely persistent synergid degeneration and elimination. These results indicate a RALF-RBOH-MitoROS-based signaling mechanism to mediate mitochondrial oxidative stress and dPCD in the fertilization process of flowering plants.

PCD is an essential physiological process that plays a vital role in the development of both animals and plants. Cells undergoing PCD display characteristic morphological and biochemical features that include shrinkage, chromatin condensation, and activation of lytic enzymes for cellular component degradation[1–3]. These degenerative alterations were not observed in the persistent synergid cell except for nuclear disintegration[15,18,19]. Due to a lack of cytological and molecular evidence, it has recently been discussed that the elimination of persistent synergid is not achieved by PCD, but by a cell fusion event followed by its nuclear disintegration in the primary endosperm[4]. In this work, we have shown that fertilization-induced accumulation of the autocrine signaling ligand ZmRALF12 correlates with mitochondrial ROS production resulting in active autophagosome production and vacuole accumulation shortly before persistent synergid elimination. Additionally, a number of senescence regulators and senescence-promoting PCD TFs were strongly activated after 18 HAP, demonstrating that stress-induced senescence occurs in persisting synergid cells. Moreover, consistent with activation of the autophagic process and vacuole accumulation, we also detected genes for key vacuolar cysteine proteases, ZmCEP1 and ZmVPEγ, which were de novo induced during PCD execution for vacuole-related corpse clearance. These

results provide direct evidence that persistent synergid cells indeed undergo a typical PCD/degeneration process shortly before cell elimination by SE fusion. After SE fusion, the PCD TFs and executors would be highly diluted in the huge primary endosperm and degraded by the endosperm factors, in a similar way to the rapid dilution of pollen tube attractants from the persistent synergid cell[15], thus enabling normal endosperm development.

Studying the elimination of the persistent synergid cell is unique as it allows to access cells undergoing dPCD at highly precise timing as degeneration of the persisting synergid cell is triggered by fertilization. In addition to the GRN controlled by MYB98 required for synergid functions before fertilization (Fig. 7a), we have shown that apparently further networks controlled by HB transcription factors are significantly downregulated shortly after fertilization (Fig. 2f), concomitant with apparent transcriptional downregulation of most highly expressed synergid-specific/predominant CRP genes with potential roles in pollen tube attraction and reception (Supplementary Data 5). In *Arabidopsis*, It has been reported that fertilized egg cells secrete endopeptidases to degrade pollen tube attractants, representing a relatively fast post-fertilization block to polytubey[58]. Our high temporal resolution cell- and stage-specific RNA-seq data imply that in addition to post-translational peptide degradation, there also exists fertilization-induced degradation of mRNAs and downregulation of their genes to reduce attraction factors. In contrast to a significant decline of synergid-specific GRNs shortly after fertilization, the activation of persistent synergid PCD execution occurs with considerable delay (i.e., after 18 HAP/~10 h after gamete fusion). Consequently, persistent synergid cells of maize remain intact for about 10 h after pollen tube perception, allowing the attraction of secondary pollen tubes for another chance of fertilization in case of fertilization failures, a mechanism referred to as fertilization recovery[11].

In conclusion, our data reveal a high-resolution atlas of timely activation and execution of dPCD and additionally provide a rich resource for future studies investigating so far unknown dPCD regulators. As further summarized in our working model (Fig. 7b, c), this study elucidates how autocrine ZmRALF12-triggered mitoROS induces dPCD, marked by the onset of oxidative stress, and which works synergistically with activated ethylene production and signaling to ensure timely dPCD execution and corpse clearance shortly before SE fusion. It will now be exciting to find out whether misexpression of ZmRALF12 is sufficient to trigger PCD also in other cell types, whether its RBOH targets are localized to mitochondrial membranes, and whether RALF12-triggered mitoROS production in synergid cells is conserved among plants. Functional studies are now necessary to elucidate the roles of previously undescribed dPCD-activation-related genes to better understand the molecular machinery responsible for driving PCD during development and reproduction in maize and other flowering plants.

## Methods

### Plant material, transgenic plants, and growth conditions
Maize (*Zea mays*) inbred line B73 was used for manual cell isolation and cell-specific RNA-seq. Transgenic line *pZmES4::ZmES4-GFP* was constructed in the previous study[21] and *pZmSC1::ZmSC1-GFP* as well as *pZmSC3::ZmSC3-GFP* transgenic lines were generated in the B104 background. To generate GFP reporter lines, gene promoter sequences (around 3.2 kbp of *ZmSC1* and 2.6 kbp of *ZmSC3*) and their respective coding sequences were amplified and cloned into the vector pTF101.1[59] upstream of GFP. The CRISPR-Cas9 technique was used to generate *zmralf12* knockout lines. Target sites were designed and cloned into the pEGCas9PUB-B vector. Constructs were transformed into B104 via *Agrobacterium*-mediated transformation (Edgene Biotech). Primer sequences used in this study are shown in Supplementary Data 10. Plants were cultivated in the greenhouses at 26 °C under illumination of 24,000 lux with 16 h light/8 h dark cycles and a relative humidity of 60%.

### Isolation of synergid cells with precise timing
Synergid cell stages were defined according to the precise time after pollination/fertilization. Synergid cells at defined stages were isolated using a protocol described previously with minor modifications[27,60]. In brief, ovules in the middle part of maize cobs were collected and sliced manually. Ovular sections containing embryo sacs were treated for 10 min with 1% cellulase R10 and 0.5% macerozyme R10 (Yakult Pharmaceutical) dissolved in 11% mannitol and 0.058% MES pH 5.8. Next, samples were extensively washed in the same solution without cell wall degrading enzyme. Microdissection of synergid/persistent synergid cells was performed with a very thin glass needle. All cells isolated from ovules were washed three times in the same solution, individually transferred to 0.5 mL RNA/DNA LoBind microcentrifuge tubes (Eppendorf), immediately frozen in liquid nitrogen, and stored at −80 °C for RNA extraction.

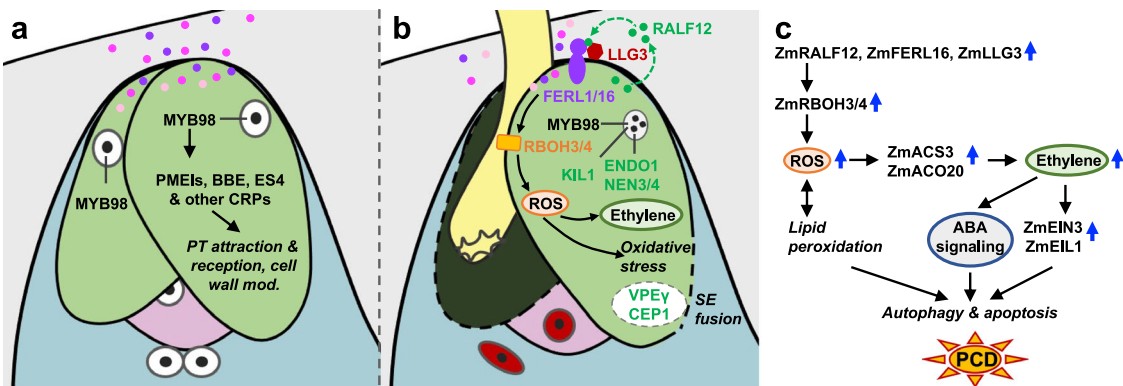

**Fig. 7 | Model of ZmMYB98 controlled synergid cell functions and subsequent ZmRALF12 induced persistent synergid PCD after fertilization. a** Before fertilization, ZmMYB98 positively regulates most highly expressed synergid-specific/predominant genes required for pollen tube attraction and reception as well as cell wall modification. **b** After successful double fertilization, ZmMYB98 and its associated gene regulatory network (GRN) are downregulated, while a GRN is activated leading to RALF12 signaling culminating in high levels of mitochondrial ROS accumulation at the filiform apparatus adjacent region, oxidative stress, mitophagy, and synergid apoptosis. Activated ethylene biosynthesis further promotes sensitized synergid cell death. RALF12-mediated mitochondrial oxidative stress and ethylene biosynthesis can individually trigger synergid cell death, while they work synergistically to efficiently activate PCD TFs and PCD executors for persistent synergid nucleus and whole-cell elimination. Vacuole rupture as well as cell intercalation mediated by VPEγ and CEP1 likely contribute to synergid-endosperm (SE) fusion. **c** Schematic model of the GRN leading to persistent synergid PCD.

## cDNA preparation and library construction for RNA-seq

RNA extraction and cDNA preparation were performed according to our previous protocol with minor modifications[22]. Briefly, mRNA was isolated from cell samples (25–28 cells for each biological replicate at each stage) using the Dynabeads mRNA DIRECT™ Micro Kit (Life Technologies). cDNA synthesis and amplification were performed using a SMART-Seq v4 Ultra Low Input RNA Kit for sequencing (Clontech). cDNA was purified using an Agencourt AMPure purification kit (Beckman Coulter). After purification and quantification, cDNA was used for library construction. RNA-seq libraries were prepared using a NEBNext Ultra DNA Library Prep Kit for Illumina (New England Biolabs) according to the manufacturer's instructions and sequenced on an Illumina Novaseq platform using paired-end 150 bp flow cells.

## RNA-seq data analysis

Clean reads of samples were prepared using Cutadapt version 1.15[61] and an in-house Perl script. Clean reads were mapped to the maize genome (AGPv5, Zm-B73-REFERENCE-NAM-5.0, http://ftp.ensemblgenomes.org/pub/plants/release-51/fasta/zea_mays/dna/) using Hisat2 version 2.0.5[62]. Gene expression levels were quantified as FPKM using featureCounts version 1.5.0-p3[63]. Differential expression analysis of stage-to-stage comparisons (three biological replicates per stage) was performed using DESeq2[64]. Genes with a $|\log_2 FC| > 1$ and false discovery rate (FDR) < 0.05 were considered significantly differentially expressed.

## Dual-luciferase reporter assay

A dual-luciferase reporter assay was performed to analyze promoter activities. To generate reporter constructs, promoters (around 2 kbp) of the 19 most highly expressed synergid-specific/predominant genes were amplified and cloned upstream of the *LUC* gene in the reporter vector pGreen II 0800-LUC[65]. The full-length coding sequence of *ZmMYB98* was cloned into pGreen II 62-SK[65] plasmid downstream of the cauliflower mosaic virus (CaMV) 35S promoter to generate an effector vector. The constructs were introduced into the *Agrobacterium tumefaciens* strain GV3101. *Agrobacterium* cells containing the reporter and effector constructs were mixed and infiltrated into tobacco leaves. After incubating for 72 h, infected areas of the leaves were collected to measure the activity of firefly luciferase (LUC) and renilla luciferase (REN) using a Dual-Luciferase Reporter Gene Assay Kit (Beyotime Biotech). Three biological replicates were performed for each experiment. Primer sequences used are listed in Supplementary Data 10.

## ROS staining and quantification

For ROS stainings, micropylar ovule regions containing embryo sacs (whole embryo sacs containing a visible filiform apparatus) or manually isolated synergid cells were dissected before pollination and at 20 HAP, respectively. A Reactive Oxygen Species Assay Kit (Beyotime Biotech) was used to detect intracellular reactive oxygen species. Samples were immersed in ROS staining solution (7.5 µM DCFH-DA in 11% mannitol solution, pH 5.8) in the dark for 10 min, and then washed three times with 11% mannitol pH 5.8. Imaging was carried out using a Leica confocal laser scanning microscope (CLSM) SP8 with an excitation wavelength of 488 nm. Laser intensities for all comparative samples were exactly the same within one experiment. ImageJ[66] was used to determine fluorescence intensity.

For ROS and Mito-Tracker double staining, cells were incubated in double staining solution (7.5 µM DCFH-DA and 20 nM Mito-Tracker Red CMXRos [Beyotime Biotech] in 11% mannitol solution, pH 5.8) in the dark for 10 min, extensively washed in 11% mannitol solution, and then imaged using a CLSM SP8 with an excitation wavelength of 488 nm (ROS) and 552 nm (Mito-Tracker). Relative fluorescent intensity of cells was measured using the LAS-X software (Leica).

## ZmRALF12 antibody preparation and immunofluorescence

The coding sequence of the predicted ZmRALF12 mature peptide[25] (amino acids 70–118) was cloned and inserted into the pET32a (Novagen) vector. Primer sequences used are listed in Supplementary Data 10. Recombinant ZmRALF12 produced in *Escherichia coli* was purified as antigen. ZmRALF12 antibodies were produced in rabbits (Proteingene Biotech). For immunofluorescence analysis, whole embryo sacs were collected and fixed in 4% paraformaldehyde for 3 h, washed three times with PBS, and incubated in PBS containing 3% BSA for 1.5 h. Embryo sacs were then washed and incubated overnight at 4 °C with an anti-ZmRALF12 antibody (1:200 dilution). Subsequently, samples were washed with PBS containing 0.1% triton and incubated for 1.5 h with a goat anti-rabbit Dylight 488-conjugated secondary antibody (Abbkine, A23220, 1:400 dilution). After extensive washing with PBS containing 0.1% triton, samples were imaged using a CLSM SP8 system (Leica) with 488 nm excitation.

## Yeast two-hybrid assay

For the yeast two-hybrid (Y2H) assay, the predicted coding sequence of the mature ZmRALF12 peptide[25] (amino acids 70–118) was cloned into the pGBKT7 vector; the coding sequences for the ZmFERL1 ectodomain (amino acids 20–443), ZmFERL16 ectodomain (amino acids 32–413) and ZmLLG3 excluding the signal peptide sequence (amino acids 25–145) were cloned into the pGADT7 vector, respectively. Both bait and prey plasmids were co-transformed into competent yeast cells AH109 and grown on a selection medium (Clontech) following the manufacturer's instructions. Primer sequences used are listed in Supplementary Data 10.

## RALF peptide preparation and treatments

*ZmRALF1*, *ZmRALF2*, and *ZmRALF12* coding sequences[25] without signal peptides were amplified and cloned into the pET-32a (Novagen) vector, respectively. The generated vector was transformed into *E. coli* strain BL21 (DE3). A list of gene-specific primers used for the above constructs is included in Supplementary Data 10. Recombinant ZmRALF1, ZmRALF2, and ZmRALF12 (RALF) were expressed according to the PET system instructions (Novagen), and purified with TALON Metal Affinity Resin (Clontech) according to the manufacturer's instructions.

The ovule culture system was optimized based on the previous report[67]. The liquid medium for maize ovule/ovule tissue culture contained the Nitsh base salts with vitamins (Coolaber), 7% (w/v) trehalose dihydrate (Aladdin), and 0.3% (w/v) PIPES-KOH (pH 5.8) (Aladdin). For RALF treatment of ovule tissues, ovular sections containing embryo sacs were incubated in a culture medium with each 10 µM RALF peptide for 30 min. After extensive wash, samples were shortly treated for 3 min with 0.5% cellulase R10 and 0.25% macerozyme R10 (Yakult Pharmaceutical) dissolved in 11% mannitol and 0.058% MES pH 5.8. Whole embryo sacs were quickly separated using a glass needle and double stained as described above.

For RALF treatment of cell samples, synergid cells from mature virgin ovules were manually isolated and incubated in solution of each 4 µM RALF peptide in 11% mannitol solution for 30 min. Cell samples were extensively washed after incubation. ROS staining or double staining procedures were immediately performed after treatment as described above.

## Analysis of mitochondrial ROS production in vitro

Synergid cells before pollination were manually isolated and collected in 11% mannitol solution. For RBOH inhibitor treatment, synergid cells were preincubated with 11% mannitol solution containing 10 µM VAS2870 (Aladdin) for 30 min before application of 4 µM RALF12 for 30 min. To examine the involvement of $Ca^{2+}$ and $K^+$ channels in ZmRALF12-triggered mitochondrial ROS production, synergid cells were preincubated with 11% mannitol solution containing 0.5 mM $LaCl_3$

(Ca$^{2+}$ channel inhibitor; Aladdin) or 0.5 mM CdCl$_2$ (K$^+$ channel inhibitor; Aladdin) for 10 min prior to application of 4 μM RALF12 for 30 min. For H$_2$O$_2$ treatment, synergid cells were incubated in 11% mannitol solution containing 30 μM H$_2$O$_2$ for 30 min. To determine the involvement of Ca$^{2+}$ channels in H$_2$O$_2$-triggered mitochondrial ROS production, synergid cells were preincubated with 11% mannitol solution containing 0.5 mM LaCl$_3$ (Aladdin) for 10 min prior to application of 30 μM H$_2$O$_2$ for 30 min. After treatment, cells were extensively washed three times with 11% mannitol solution and double stained with DCFH-DA and Mito-Tracker Red as described above.

## MDC staining
Synergid cells and persistent synergid cells at 24 HAP were manually isolated and collected in 11% mannitol solution. For RALF treatment assays, ovular sections containing embryo sacs before pollination were incubated in a culture medium with each 10 μM RALF peptide at 28 °C for 6 h. Synergid cells were then manually isolated and collected in 11% mannitol solution. Autophagic structures in cells were visualized by fluorescent monodansylcadaverine (MDC) staining using an Autophagy Staining Assay Kit with MDC (Beyotime Biotech) following the manufacturer's instructions. Images were captured with a Leica TCS SP8 CLSM with 405 nm excitation.

## ROS scavenger treatment and analysis
Ovules were harvested at 16 HAP and cultured in a liquid medium with or without ROS scavenger CuCl$_2$ (5 mM), respectively. After cultivation at 28 °C for 6 h, whole embryo sacs were quickly dissected and double stained as described above. After cultivation at 28 °C for 12 h, embryo sacs were isolated and observed to determine the frequency of persistent synergid cell-containing embryo sacs.

## Ethylene treatment and analysis
For ethylene treatment, the middle parts of unpollinated cobs were placed in 100 ml sealed glass bottles to incubate with gaseous ethylene released from ethephon (Aladin Biotech). 0.1 g ethephon was dissolved in 1 ml sterile pure water in a small glass dish beside dissected cobs. After 30 h, cobs were collected and treated for CLSM analysis according to a protocol described previously with some modifications[68]. Briefly, ovules were dissected with two longitudinal sections along the silk axis. Ovule pieces containing embryo sacs were fixed in a solution of 4% glutaraldehyde in 12.5 mM cacodylate (pH 6.9) for 2 h at room temperature and then at 4 °C overnight. Following fixation, samples were dehydrated by sequential treatment with 10%, 20%, 40%, 60%, 85%, 95%, and 100% ethanol for 30 min at each step. After dehydration, samples were cleared in a 2:1 mixture of benzyl benzoate:benzyl alcohol for 20 min. Cleared sections were analyzed using a Leica TCS SP8 CLSM with an excitation wavelength of 488 nm.

## Confocal microscopy analysis of synergid degeneration processes
To examine synergid cell morphology and its removal at defined developmental stages, ovules were collected and manually dissected along the silk axis. Ovular sections containing embryo sacs were fixed and analyzed as described above for ethylene-treated samples. To visualize GFP and SE fusion dynamics, *pZmES4::ZmES4-GFP* and *pZmSC1::ZmSC1-GFP* cobs were harvested at indicated stages, ovules were dissected and kept in 11% mannitol solution. Microscopy was performed on the Leica TCS SP8 CLSM with 488 nm excitation. Laser intensities for all comparative samples were exactly the same within one experiment.

## Statistics and reproducibility
Statistical methods were not used to predetermine sample size and no data were excluded from the analyzes. The experiments were not randomized. The Investigators were not blinded to allocation during experiments and outcome assessment. As many replicates as possible were done for the papers. Most key data such as RALF peptide treatments, ethylene treatments, and ROS scavenger applications have generally been repeated at least three times. See the figure legends for details. The deviations between biological replicates are small and are thus highly reproducible and sufficient to draw solid conclusions. The data analysis and presentation are done with Graphpad Prism 10.1.0.

## Reporting summary
Further information on research design is available in the Nature Portfolio Reporting Summary linked to this article.

## Data availability
The RNA-seq data generated in this study have been deposited in the NCBI Gene Expression Omnibus (GEO) under accession code GSE283809. Previously published RNA-seq data (GEO accession code GSE98379) were also used in the present study. The core data of transcriptomics in the paper are also provided in the supplementary data file. Source data are provided in this paper.

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

## Acknowledgements

We would like to thank Mengxiang Sun and Xiongbo Peng (Wuhan University) for assistance in teaching manual cell isolation techniques. Wenliang Xu (Central China Normal University) is acknowledged for help with the dual-luciferase assay system and Armin Hildebrand (University of Regensburg) for plant care. This work was supported by grants from the National Natural Science Foundation of China (31970342 to J.C.), the Major Project of Hubei Hongshan Laboratory (2022hszd017 to J.C.), the Ten Thousand Talents Program for Young Talents (to J.C.), the Alexander von Humboldt Foundation (to X.Z.) and the German Research Foundation (DFG) via research unit FOR 5098 ICIPS (to T.D.).

## Author contributions

J.C. and T.D. conceptualized the study, designed the research plan, interpreted the results, and wrote the manuscript with input from all authors. H.W., J.W., and J.C. collected cell samples, J.C. analyzed transcriptomic data with support from T.D., L.H., and S.H.; H.W., J.W., J.C., X.Z., W.Q, H.F., and S.W. performed the experiments. All authors contributed to data collection and discussion, presentation, and finalizing the manuscript.

## Funding

## Competing interests

The authors declare no competing interests.
