## [Transparent Peer Review file · Nature Communications]

Fertilization-induced synergid cell death by RALF12-triggered ROS production and ethylene signaling

Corresponding Author: Professor Thomas Dresselhaus

Version 0:

Reviewer comments:

Reviewer #1

(Remarks to the Author)

Understanding the mechanisms that control the reproduction of plants is not only important for basic research, but can also have far-reaching implications for applied sciences. In this beautifully written paper, the authors decipher the developmental processes of the non-receptive synergid cell throughout fertilization. The manuscript presents an impressively detailed transcriptome analysis, covering the phase from the stage where the synergids primarily act as pollen-tube attracting secretory gland cells to nuclear disintegration and cell fusion, which manifests after fertilization. This includes significant developmental milestones such as increased ROS production, initiation of ethylene signaling, disintegration of the synergid nucleus, and ultimately, pSC-ES fusion. The data analysis and statistical methods are robust, and the majority of the authors' conclusions are well substantiated by the provided data.

In support of a nuclear disintegration process, the authors show that following fertilization several synergid specific TFs are down regulated, including the previously described MYB98. In addition, they show that MYB98 induces the promoters of several synergid specific secretory proteins cementing its established pivotal role in defining synergid cell identity.

The authors provide conclusive evidence that RALF 12 which is involved in an autocrine signalling pathway leads to the accumulation of ROS in mitochondria. It was previously shown that synergid located FERONIA/ANJ/HERK receptor interact with different RALFs released from pollen tubes to facilitate ROS release, pollen tube reception and the degeneration of receptive synergid1-3 (some of the info is missing in the paper). However, the information that RALF12 is upregulated in maize synergid cells to contribute to ROS induction and PCD is new.

Additionally, the authors demonstrate the upregulation of ethylene biosynthesis genes (ZmACS3, ACO5, ACO20) and illustrate that treating maize embryo sacs with ethylene results in the specific degeneration of synergids, while other cells within the embryo sac remain unaffected. The pivotal role of ethylene signaling and the heightened susceptibility of synergid cells compared to other cells within the FG have been previously established: injection of the ethylene precursor ACC into the central cell of the FG leads to the disintegration of synergid nuclei, with no impact on other cells. Moreover, constitutively active ethylene response in *ctr1* mutants triggers the disintegration of synergids while leaving other cells intact⁴. This information should be added to this manuscript. The finding that synergids undergo premature disintegration after direct ethylene application offers a valuable piece to the puzzle, even though it does not resolve the matter, as previous research has shown that the depletion of ethylene does not impact synergid programmed cell death (PCD)^{5,6}. Finally, the authors show that a battery of PCD-associated TFs become active and the vacuolated remnants of the persistent synergids fuse with the endosperm 26-30 HAP.

Using state of the art and in parts even explorative methods, this paper contributes new information to several known pathways involved in the development and disintegration of synergids. However, it does not conceptually advance our understanding of the underlying processes. Still the meticulous yet powerful cell isolation technique, coupled with high temporal resolution and in-depth transcriptome analysis in a crop plant, which inherently poses technical challenges, presents a very valuable resource for the scientific community interested in plant reproduction.

Major concern:

As already mentioned, the introduction and discussion need to be amended to better reflect the current state of the art.

Minor comments:

Line 49-50: Notably, polyspermy leads to lethal genome imbalance and chromosome segregation defects during embryo development¹³: This statement holds true for animals and the citation provided is outdated for the plant kingdom. Plant sperm does not exhibit centrioles and chromosome segregation defects are tolerated in plants.

The word "notably" (l186) is too strong given that it is not surprising that TF transcript levels go down in a disintegrating synergid.

The term "degeneration" for the non-receptive synergid is misleading as only the nucleus disintegrates while the remainder of the cell fuses with the CC.

Typos:

Line 323: ... bot not

Line 350: Strikingly

Line 427: unkques

Line 454: consertvedf

1 Zhong, S. et al. RALF peptide signaling controls the polytubey block in Arabidopsis. *Science* 375, 290-296, doi:doi:10.1126/science.abl4683 (2022).

2 Duan, Q. et al. Reactive oxygen species mediate pollen tube rupture to release sperm for fertilization in Arabidopsis. *Nat Commun* 5, 3129, doi:10.1038/ncomms4129 (2014).

3 Völz, R., Harris, W., Hirt, H. & Lee, Y.-H. ROS homeostasis mediated by MPK4 and SUMM2 determines synergid cell death. *Nature Communications* 13, doi:10.1038/s41467-022-29373-7 (2022).

4 Völz, R., Heydlauff, J., Ripper, D., von Lyncker, L. & Groß-Hardt, R. Ethylene signaling is required for synergid degeneration and the establishment of a pollen tube block. *Dev Cell* 25, 310-316, doi:10.1016/j.devcel.2013.04.001 (2013).

5 Li, W. et al. Lack of ethylene does not affect reproductive success and synergid cell death in Arabidopsis. *Mol Plant* 15, 354-362, doi:10.1016/j.molp.2021.11.001 (2022).

6 Heydlauff, J. et al. Dual and opposing roles of EIN3 reveal a generation conflict during seed growth. *Mol Plant* 15, 363-371, doi:10.1016/j.molp.2021.11.015 (2022).

Reviewer #2

(Remarks to the Author)

In this manuscript, Chen and colleagues examine the process around the loss of the second synergid cell after fertilisation. Studying this process is highly complex given that a single cell (or a group of a few cells at most), is being studied, the cells are deeply embedded in maternal tissues, and the process is quite rapid. To study this in maize, which has more limited genetic resources than Arabidopsis, adds to the difficulties of this research. The progress the team have made in understanding this critical step in reproduction is therefore commendable. That being said, there are a number of places where I think this manuscript should be improved.

Major points:

This paper implies that ZmRALF12 is the peptide responsible for induction of ROS in the persistent synergid, leading to its fusion with the central cell and hence loss of function. While the data on ZmRALF12's ability to induce ROS production is pretty clear, that this is its *in vitro* function, and that ZmRALF12 is necessary and sufficient for pSC degeneration is less convincing. I think two additional experiments at least are required to strengthen this claim:

- 1) Expression of ZmRALF12-GFP (and possibly also pRALF12::GUS) to demonstrate that it is indeed expressed in the persistent synergid at the correct stage, and to show where this peptide is localised.
- 2) Knockout/down of ZmRALF12 to show that ROS does not then accumulate at the expected timepoint after pollination/fertilisation.

A number of central conclusions from this paper are based primarily on data from protoplasts of synergid cells, where protoplasting may induce cell stress responses. How closely these results reflect what happens in planta is a little difficult to assess without follow-up in planta experiments showing e.g. RALF12 responses and mitochondria localisation in planta.

In experiments looking at localisation of ROS and mitochondria in dissected persistent synergids, it's not possible for an untrained eye to see where the filiform apparatus is located. Co-localisation (line 280) should therefore be demonstrated by experiments that show ROS or mitochondria location alongside a known reporter for the filiform apparatus.

I found the section on ethylene responses vs ethylene production a little hard to follow. The authors hypothesised that increased ethylene production might trigger cell death, but then tested responses to exogenous ethylene. While changing ethylene production levels in maize synergids may be difficult, this would be the logical experiment given the stated hypothesis. Alternatively, some rewording of this section may be sufficient to make it easier to follow the logic.

Minor points:

1. In the introduction, further explanation of the steps mentioned on line 56 would be helpful for a non-specialist reader.
2. Addition of a timelapse video or video showing what happens with the synergids after pollen tube reception would be a

very useful addition for non-specialists.

3. The words used to describe what happens to the persistent synergid vary, which would make the manuscript less accessible to some readers: elimination, removal, deletion, apoptosis, degeneration, programmed cell death. I think it would be better to restrict this to one or two terms which are clearly defined early in the manuscript and consistently used throughout.
4. Receptive synergid, degenerated receptive synergid, and degenerated synergid seem to be used as synonyms. It might be clearer to explain how these terms are used in the introduction and ensure consistent use thereafter.
5. Line 54 – The authors mention that whether the persistent synergid undergoes PCD is disputed. I feel this could have been more deeply discussed in the conclusion section.
6. Legend of Figure S1 could benefit from mention of the B73 maize line.
7. Line 105 – typo. Should be 'collected' rather than 'collect'
8. How many cells were combined for each biological replicate of RNA-seq? Or is this scRNA-seq?
9. Line 123-125 – the authors state that ZmES4 goes down after fertilisation since it is required for pollen tube burst. However, it then goes up again from pSC18. Why might this occur?
10. Where does the egg cell/zygote RNA-seq data come from? Was it also generated as part of this study? Either way, the source of this comparative data could be clearer in the manuscript.
11. Figure 1 – I'm not convinced that 'High-precision' is needed or correct in the figure title. I think a more informative phrase could be used here, or the words simply omitted.
12. Figure 1 j and k – Use of different arrow types to indicate the pollen tube entrance and sperm cell release regions would be better.
13. Figure 1m – some stats here to show significant differences between time points would be appropriate.
14. In Figure S2, the correlation between RNA-seq samples looks to be lower than the average R-squared given in the main text of 0.93. Obviously self-correlations should be excluded from this calculation, and 11 of the 66 values lie between 0.93 and 0.959 while 55 are between 0.769 and 0.93.
15. Line 145 – need to specify what at which time point you analysed gene expression patterns.
16. Line 149 and/or supplementary data 3 – need to explain what is meant by TOP200 category.
17. Line 169 – typo. Should be 'which' rather than 'with'
18. Figure 2f, 3e, 3f, 5a – statistics needed.
19. Figure 3h – inclusion of a negative control is needed
20. Line 818 – fluorescence intensity relative to what?
21. Line 232-4 and Line 255– grammar needs fixing.
22. Figure 6 i, j, and l – statistics needed. Also, it would be better to show individual data points and keep presentation of expression data consistent between figures (also Fig S8).
23. Line 323 – typo. Should be 'but' rather than 'bot'
24. Line 387-9 – phrasing could be improved.
25. Line 427 – grammar
26. Line 432 – it would be helpful to refer to the figure which demonstrates the point you make here.
27. Line 454 – spelling.
28. Line 507 – should include *Agrobacterium* species and strain.
29. Figure S9c and d – should the asterisk be lower down? Looks to be on the dSC based on other figures, rather than on the remnants of the pSC.
30. Figure S5c – should include number of replicates
31. Given you propose that PCD precedes fusion of the persistent synergid with the endosperm, does that imply that a dead cell fuses with a living cell? How does the endosperm recover from partial plasmolysis? Could expression of some previously uncharacterised genes during PCD in the synergid be to limit the effects of PCD on the endosperm after fusion? Perhaps some speculation on this could be included in the discussion.
32. Supplementary data 9 – what is meant by 'de novo' here? The normal meaning of the phrase doesn't seem quite appropriate here as many of the genes listed do have a low expression level prior to PCD.
33. Many genes are referred to as being 'synergid-specific' when either they are also expressed elsewhere according to the provided supplementary data files, and/or there is insufficient data to conclude that these genes are ONLY expressed in the synergids and not anywhere else in the plant. Maybe 'synergid-expressed' or 'synergid-enriched' would be more accurate as a descriptor?
34. Read order/labelling according to citation order could be improved in some of the figures.
35. Supplementary data figures should be cited in strict numerical order.

Reviewer #3

(Remarks to the Author)

Summary

This study used isolated synergid protoplasts from maize ovules at different stages post-fertilization to investigate the regulators of persistent synergid degradation. An accumulation of ROS was observed in persistent synergids 20 hours after pollination. RNA-Seq data indicated upregulation of a RALF signaling pathway and two NADPH Oxidases. Exogenous application of RALF 12, but not RALF1 or 2, to synergids induced ROS accumulation suggesting this peptide might regulate ROS accumulation during persistent synergid degradation. Mitochondrial association with the filiform apparatus was observed along with autophagosome accumulation 24 hours after pollination. Ethylene treatment could induce synergid degradation in ovules without affecting the egg or central cell, suggesting that synergids are primed for ethylene-mediated degradation. This study also identified new transcription factors that function as regulators of programmed cell death and are upregulated during persistent synergid degradation.

Major Strengths

The authors were able to isolate synergids to use for RNA-Seq and various treatments at different stages post-fertilization. The transcript data for persistent synergids at various time points allows for a dissection of how transcriptional regulation of genes changes from fertilization through the degradation of the persistent synergid. The data in the paper are well-presented and follow a clear logic to dissect the possible regulation of persistent synergid elimination. The most exciting result was that RALF12 was identified as the RALF family member upregulated during synergid degradation. RALF12 was able to induce ROS accumulation in isolated synergids *in vitro*. Previous studies investigating the role of ethylene in synergid degradation were done in *Arabidopsis*. The authors used exogenous application of ethylene to show that ethylene was sufficient to induce synergid degradation in maize ovules *in vitro* and that ethylene related genes were upregulated 24 hours after pollination, suggesting it might play a role in maize synergid degradation. 6 NAC transcription factors previously shown to induce programmed cell death are upregulated in synergids 24 hours after pollination and are candidate regulators of persistent synergid degradation. Overall, the manuscript makes and tests some interesting hypotheses about how the persistent synergid degrades following double fertilization.

Major Weaknesses

The authors didn't investigate whether making synergids protoplasts in order to isolate them affects the functionality of these cells. In most of the images of isolated synergid protoplasts, it appears that the synergids become round and the filiform apparatus does not keep its shape. For example, in Fig 3, some of the isolated synergids are completely round and it is difficult to see something that looks like a filiform apparatus. The authors should include information on how the filiform apparatus is identified in the protoplasts since many of their conclusions are based on identifying higher signal at the presumed filiform apparatus. Analysis of synergid protoplasts made from the pSC3::SC3 line (or another filiform apparatus-specific transgene) at the different time points would help define what happens to the filiform apparatus in protoplasts and could help refine the conclusions from the microscopy experiments.

Line 435 suggests that RALF12 specifically triggers mitochondrial ROS production in the filiform apparatus. However, the data presented in this paper does not fully support that model, especially since, based on many studies in *Arabidopsis*, the RALF peptides are expected to increase ROS through FERL-RLK signaling that upregulates the activity of plasma membrane associated RBOHs. Mitochondrial ROS is from a different source. The authors should discuss how the two NADPH Oxidases (RBOH3 and RBOH4) that are upregulated in the persistent synergid and presumably plasma-membrane localized might be involved in the mitochondrial ROS that accumulates in persistent synergids during the death process. Furthermore, it is unclear whether mitochondria associate with the filiform apparatus constitutively or if that is in fact induced by RALF12 and how the mitochondria contribute to the increased ROS. A OHAP control should be included in Fig. 4 to show mitochondrial localization prior to fertilization or RALF12 treatment in both intact synergids and in synergid protoplasts.

This paper makes conclusions about the processes regulating the death of the persistent synergid from the RNA-Seq data. Many of the claims such as Ca²⁺ causing a shift from redox balance to oxidative stress were not proven experimentally. It's possible that post-transcriptional or translational regulation might show that protein activity doesn't correlate with the increased transcript levels of certain genes. In light of this potential complication, the authors should be more cautious in their interpretation of this transcriptome data.

Lines 313-317 suggest that RALF12 triggers mitochondrial ROS production and leads to programmed cell death and autophagy. The current data presented only shows a link between ROS and autophagy through upregulated genes, but does not show a direct correlation. Imaging with MDC staining in RALF12 treated and untreated synergids to determine if the ROS induced by RALF12 can also induce autophagosome production would more directly show whether programmed cell death is triggered by RALF12.

Minor Weaknesses

Statistical comparison for Figure 2F is missing. In the text (line 186) it suggests that all the transcription factors included decrease in the pSC compared to the control, but there isn't a statistical test done to demonstrate this and not all the genes look like they have reduced expression in the pSC.

Nearly all figure panels depicting FPKM for specific genes in SC, pSC12, pSC18, and pSC24 are missing statistical comparisons (Figures 1M, 3E, 3F, 5A, supplemental figures 6, 7, 8).

In Figure 3H and supplemental figure 5 the synergid cells start looking deformed at 30 minutes or a concentration of 2µM RALF12. Does the deformation of the synergid have any biological relevance? Could the same dosage/time series depicted in supplemental figure 5 be done with RALF1 or RALF2 to dissect whether it is the ROS induced by RALF12 that causes this change in morphology or if the presence of a RALF in the absence of ROS would also cause the same change.

Reviewer #4

(Remarks to the Author)

Reviewer #5

(Remarks to the Author)

Version 1:

Reviewer comments:

Reviewer #1

(Remarks to the Author)

The revised manuscript 'Fertilization-induced synergid cell death by RALF12-triggered ROS production and ethylene signaling' is clearly improved over the earlier version. While the high-resolution RNA-seq data provides clear evidence for the activation of molecular programs characteristic for programmed cell death (PCD) in persistent synergids, PCD still is no appropriate term, as a hallmark of PCD is the disintegration of DNA and the nucleus(1-3). Accordingly, the authors should refrain from using this term like previous authors did (4).

What (still) stands out negatively about this paper is its disregard for highly relevant existing literature and data. This is not only misleading for reviewers, it also takes credit away from others' work: While the authors now integrate the requested publications regarding RALF and ethylene signaling the manuscript does not provide/discuss information on the extensive work previously published on the role of MYB98 during synergid disintegration in Arabidopsis and the existing knowledge on differential gene regulation by MYB98 during this process. Currently the MS reads as if the role of MYB98 for synergid disintegration was first discovered by the Dresselhaus lab. This needs correction.

A key contribution of this study lies in demonstrating the role of RALF12 in an autocrine signaling pathway that triggers intracellular (mitochondrial) ROS production, which is distinct from the previously described apoplastic ROS burst in receptive synergids. The additional in vivo experiments, including the generation of a *ralf12* CRISPR null mutant, effectively substantiate the significance of RALF12-induced mitochondrial ROS. Further, the authors highlight how ethylene signaling acts synergistically with this RALF12-dependent mechanism to drive the disintegration of the persistent synergid.

In their revised version, the authors also clarify previously raised questions about the timeline of synergid disintegration.

The authors have made an effort to address the reviewer's concern by adopting the term 'synergid-specific/predominant.' However, the use of both terms can affect readability and has been applied somewhat inconsistently (e.g., in line 200: 'the most strongly expressed synergid-specific TFs').

1 Fernández-Lázaro, D., Sanz, B. & Seco-Calvo, J. Mechanisms of programmed cell death: structural and functional pathways. A narrative review. *Investigación Clínica* (2024). <https://doi.org/10.54817/ic.v65n2a09>

2 Ren, H. et al. Calcium Signaling in Plant Programmed Cell Death. *Cells* 10 (2021). <https://doi.org/10.3390/cells10051089>

3 Huysmans, M., Lema A, S., Coll, N. S. & Nowack, M. K. Dying two deaths — programmed cell death regulation in development and disease. *Current Opinion in Plant Biology* 35, 37-44 (2017). <https://doi.org/https://doi.org/10.1016/j.pbi.2016.11.005>

4 Maruyama, D. & Higashiyama, T. The end of temptation: the elimination of persistent synergid cell identity. *Current Opinion in Plant Biology* 34, 122-126 (2016). <https://doi.org/https://doi.org/10.1016/j.pbi.2016.10.011>

Reviewer #2

(Remarks to the Author)

This is a revised version of the manuscript by Chen and colleagues on maize synergid cell death being triggered by ZmRALF12. The authors have responded thoroughly to all three reviewers, substantially strengthening and revising the manuscript in response to our many suggestions. This has clearly been quite a lot of work for which the authors should be commended. I believe the manuscript is now appropriate for publication.

Reviewer #3

(Remarks to the Author)

The revised manuscript is greatly improved due to the addition of experiments that helped the authors test the hypotheses generated based on their RNAseq data from unpollinated synergids and persistent synergids at various time points after pollination. In particular, the ROS experiments helped define the role of ROS in the persistent synergid. The link between RboH- and mitochondrial generation of ROS is intriguing and provides a new model for how RALF signaling promotes programmed cell death. The additional RALF12 crispr and immunolocalization data also strengthened the conclusion that RALF12 is involved in regulating persistent synergid cell death. I'm satisfied with the authors' responses to my critiques on the first version of the manuscript. I think that they were also able to adequately respond to the other reviewers' concerns. Minor suggestions:

Lines 534: Notably, we show that in consistence with XX” should be changed to “Consistent with XX,”
Line 563: “Moreover, in consistence with activation...” should be “Consistent with activation...,”
Figure 2 legend: please note in the legend that the data in the first pie chart in panel a is synergids before pollination
Figure 4: The label in panel 4K should be ZmRALF2
Figure 6: Change the d-1 label so that it is all on the same line

Reviewer #5

(Remarks to the Author)

Manuscript NCOMMS-24-12001

Title: “Fertilization-induced synergid cell death by RALF12-triggered ROS production and ethylene signaling” by Chen et al.

Point-by-Point Response to the Reviewer’s Comments

We sincerely appreciate the reviewers for their careful reading of our manuscript, their insightful comments, and especially for their constructive suggestions which have greatly helped us to improve the manuscript.

Following is a point-by-point summary of the changes made in the manuscript to address the points raised by the reviewers. The cell- and stage-specific RNA-seq data have been uploaded to GEO (<https://www.ncbi.nlm.nih.gov/geo/>) and will be available under accession number GSE283809. The secure token “ijunssucjnqzjir” has been created to allow review of the RNA-seq data.

Reviewer #1 (Remarks to the Author):

Understanding the mechanisms that control the reproduction of plants is not only important for basic research, but can also have far-reaching implications for applied sciences. In this beautifully written paper, the authors decipher the developmental processes of the non-receptive synergid cell throughout fertilization. The manuscript presents an impressively detailed transcriptome analysis, covering the phase from the stage where the synergids primarily act as pollen-tube attracting secretory gland cells to nuclear disintegration and cell fusion, which manifests after fertilization. This includes significant developmental milestones such as increased ROS production, initiation of ethylene signaling, disintegration of the synergid nucleus, and ultimately, pSC-ES fusion. The data analysis and statistical methods are robust, and the majority of the authors' conclusions are well substantiated by the provided data.

In support of a nuclear disintegration process, the authors show that following fertilization several synergid specific TFs are down regulated, including the previously described MYB98. In addition, they show that MYB98 induces the promoters of several synergid specific secretory proteins cementing its established pivotal role in defining synergid cell identity.

The authors provide conclusive evidence that RALF 12 which is involved in an autocrine signalling pathway leads to the accumulation of ROS in mitochondria. It was previously shown that synergid located FERONIA/ANJ/HERK receptor interact with different RALFs released from pollen tubes to facilitate ROS release, pollen tube reception and the degeneration of receptive synergid1-3 (some of the info is missing in the paper). However, the information that RALF12 is upregulated in maize synergid cells to contribute to ROS induction and PCD is new.

Additionally, the authors demonstrate the upregulation of ethylene biosynthesis genes (ZmACS3, ACO5, ACO20) and illustrate that treating maize embryo sacs with ethylene results in the specific degeneration of synergids, while other cells within the embryo sac remain unaffected. The pivotal role of ethylene signaling and the heightened susceptibility of synergid cells compared to other cells within the FG have been previously established: injection of the ethylene precursor ACC into the central cell of the FG leads to the disintegration of synergid nuclei, with no impact on other cells. Moreover, constitutively active ethylene response in ctr1 mutants triggers the disintegration of synergids while leaving other cells intact⁴. This information should be added to this manuscript. The finding that synergids undergo premature disintegration after direct ethylene application offers a valuable piece to the puzzle, even though it does not resolve the matter, as previous research has shown that the depletion of ethylene does not impact synergid programmed cell death (PCD)^{5,6}. Finally, the authors show that a battery of PCD-associated TFs become active and the vacuolated remnants of the persistent synergids fuse with the endosperm 26-30 HAP.

Using state of the art and in parts even explorative methods, this paper contributes new information to several known pathways involved in the development and disintegration of

synergids. However, it does not conceptually advance our understanding of the underlying processes. Still the meticulous yet powerful cell isolation technique, coupled with high temporal resolution and in-depth transcriptome analysis in a crop plant, which inherently poses technical challenges, presents a very valuable resource for the scientific community interested in plant reproduction.

Response: We thank the reviewer for the valuable comments and support. The last paragraph excellently summarizes the advancement. We agree, that some of the individual novel findings do not represent a conceptually advance as it was shown in the model plant Arabidopsis that among others (i) FERONIA (FER) controls high levels of apoplastic ROS production at the entrance of the female gametophyte (i.e., the filiform apparatus region) to induce pollen tube burst and sperm cell release (Duan et al. 2014. *Nat. Commun.*, **5**: 3219), that (ii) RALF peptides released from the pollen tube interact with synergid-located FER/ANJ/HERK receptor kinases to control pollen tube reception and rupture (Zhong et al. 2022. *Science*, **375**: 290-296), that (iii) MITOGEN-ACTIVATED PROTEIN KINASE 4 (MPK4) prevents FER-controlled ROS accumulation and premature synergid cell death (Völz et al. 2022. *Nat. Commun.*, **13**: 1746), that (iv) ethylene signaling, but not its production, leads to premature synergid disintegration, with the neighboring egg cell unaffected (Völz et al. 2013. *Dev. Cell*, **25**: 310-316; Li et al. 2022. *Mol. Plant*, **15**: 354-362) and that (v) synergid-endosperm fusion (SE) occurs (Maruyama et al., 2015. *Cell*. **161**: 907-818). These studies provided significant insights into the mechanism of receptive synergid disintegration during pollen tube reception, but raised also the important issue regarding how persistent synergid cell death is timely induced and executed, and whether above findings are conserved in other plants.

Our study shows among many individual aspects that dPCD indeed occurs in persistent synergid cells of plants (see also comment below) and confirms its final elimination via SE in another – very distant – plant species. Moreover, we elucidated a novel fertilization-induced autocrine – not paracrine – RALF-FERL-ROS pathway mediating among others especially mitochondrial – not apoplastic – oxidative stress and autophagy, and which works synergistically with activated ethylene production and signaling, ensuring timely dPCD of the persistent synergid cell. We consider these findings also as a conceptual advance.

Major concern:

As already mentioned, the introduction and discussion need to be amended to better reflect the current state of the art.

Response: We thank the reviewer very much for the suggestion. We have modified and updated the Introduction accordingly and elaborated a more comprehensive Discussion. Please see the detailed edits in the Introduction and Discussion sections of the revised manuscript that also includes all references listed below.

Minor comments:

Line 49-50: Notably, polyspermy leads to lethal genome imbalance and chromosome segregation defects during embryo development¹³: This statement holds true for animals and the citation provided is outdated for the plant kingdom. Plant sperm does not exhibit centrioles and chromosome segregation defects are tolerated in plants.

Response: We thank the reviewer for this comment. We have modified/generalized the sentence stating that “In flowering plants, the attraction of supernumerary pollen tubes is avoided to prevent polyspermy and to ensure chromosomal balance and progeny health.”

The word "notably" (l186) is too strong given that it is not surprising that TF transcript levels go down in a disintegrating synergid.

Response: We thank the reviewer for the comment. In our opinion it can be discussed whether the word “notably” is too strong or not and whether it is expected that TF transcripts go down shortly after fertilization (12 HAP, thus around 4hs after fertilization) in the persisting synergid or not. We should emphasize that it has never been investigated before; transcripts of all abundant TFs could also have remained stable or even showing higher expression until persistent synergid PCD is activated with significant delay after fertilization – we found this

noteworthy as it indicates activation degradation after fertilization (about which we don't speculate, but it is very likely); therefore we thought to use a strong word. Additionally, by comparing the transcript levels of the five most highly expressed synergid-specific TFs, only three of them show a second apparent reduction after 18 HAP during synergid disintegration. Thus, for a subset of these TFs, the decline trends are not perfectly correlated with synergid disintegration, but first with the fertilization process supporting above hypothesis. We tend to think that the apparent decline of synergid-specific GRN occurs much earlier than the activation of persistent synergid PCD. In order to make it clearer, we have added statistical comparisons to Fig. 2f, edited the sentence accordingly, and discussed the different timing thoroughly in the Discussion section of the revised manuscript.

The term "degeneration" for the non-receptive synergid is misleading as only the nucleus disintegrates while the remainder of the cell fuses with the CC.

Response: We thank the reviewer for this comment and agree that we should have written the content more clearly. It was a long-standing question whether the persistent synergid undergoes typical degeneration/PCD, as degenerative alteration was not observed in the persistent synergid except for nucleus disintegration (Maruyama et al. 2015. *Cell*, **161**: 907-918). Due to a lack of cytological and molecular evidence, it was questioned whether the elimination of persistent synergid is achieved by PCD at all (Xie et al. 2022. *Curr. Opin. Plant Biol.*, **69**: 102271). We provided several lines of evidence that the persistent synergid (of maize) undergoes degeneration. First, in addition to nuclear disintegration, we showed that the fertilization-induced autocrine signal ligand ZmRALF12 specifically triggers mitochondrial ROS production and mediates oxidative stress, leading to active autophagosome production and vacuole accumulation shortly before synergid-endosperm (SE) fusion (Fig. 4f–k). Second, a number of senescence regulators and senescence-promoting PCD TFs were strongly activated after 18 HAP, demonstrating that stress-induced senescence occurs in the persistent synergid cell before its elimination (Fig. 6i, k, l). Third, being consistent with a clear activation of the autophagic process and vacuole accumulation, key vacuolar cysteine proteases (i.e., ZmCEP1 and ZmVPEγ) have been *de novo* induced during PCD execution for vacuole-related corpse clearance shortly before SE fusion (Fig. 6j). Thus, both the cell- and stage-specific RNA-seq data and cytological observations provide direct supporting evidence for the occurrence of persistent synergid degeneration. We apologize for any confusion in presenting persistent synergid degeneration, and have now elucidated it more thoroughly in the Discussion section to enhance the clarity of our presentation.

Typos:

Line 323: ... bot not

Line 350: Strikingly

Line 427: unkques

Line 454: consertvedf

Response: We have carefully revised the whole text, and highlighted the corrections including these in the revised manuscript.

1 Zhong, S. et al. RALF peptide signaling controls the polytubey block in Arabidopsis. *Science* 375, 290-296, doi:doi:10.1126/science.abl4683 (2022).

2 Duan, Q. et al. Reactive oxygen species mediate pollen tube rupture to release sperm for fertilization in Arabidopsis. *Nat Commun* 5, 3129, doi:10.1038/ncomms4129 (2014).

3 Völz, R., Harris, W., Hirt, H. & Lee, Y.-H. ROS homeostasis mediated by MPK4 and SUMM2 determines synergid cell death. *Nature Communications* 13, doi:10.1038/s41467-022-29373-7 (2022).

4 Völz, R., Heydlauff, J., Ripper, D., von Lyncker, L. & Groß-Hardt, R. Ethylene signaling is required for synergid degeneration and the establishment of a pollen tube block. *Dev Cell* 25, 310-316, doi:10.1016/j.devcel.2013.04.001 (2013).

5 Li, W. et al. Lack of ethylene does not affect reproductive success and synergid cell death in Arabidopsis. *Mol Plant* 15, 354-362, doi:10.1016/j.molp.2021.11.001 (2022).

6 Heydlauff, J. et al. Dual and opposing roles of EIN3 reveal a generation conflict during seed growth. *Mol Plant* 15, 363-371, doi:10.1016/j.molp.2021.11.015 (2022).

Response: We thank the reviewer again for providing these valuable references and have cited all of them with more detailed descriptions in the revised manuscript.

Reviewer #2 (Remarks to the Author):

In this manuscript, Chen and colleagues examine the process around the loss of the second synergid cell after fertilisation. Studying this process is highly complex given that a single cell (or a group of a few cells at most), is being studied, the cells are deeply embedded in maternal tissues, and the process is quite rapid. To study this in maize, which has more limited genetic resources than Arabidopsis, adds to the difficulties of this research. The progress the team have made in understanding this critical step in reproduction is therefore commendable. That being said, there are a number of places where I think this manuscript should be improved.

Major points:

This paper implies that ZmRALF12 is the peptide responsible for induction of ROS in the persistent synergid, leading to its fusion with the central cell and hence loss of function. While the data on ZmRALF12's ability to induce ROS production is pretty clear, that this is its *in vitro* function, and that ZmRALF12 is necessary and sufficient for pSC degeneration is less convincing. I think two additional experiments at least are required to strengthen this claim: 1) Expression of ZmRALF12-GFP (and possibly also pRALF12::GUS) to demonstrate that it is indeed expressed in the persistent synergid at the correct stage, and to show where this peptide is localised.

Response: We thank the reviewer for the helpful comments and suggestions. To detect ZmRALF12 peptides *in planta*, we have now performed immunological analyses using antibodies against ZmRALF12. Consistent with the cell- and stage-specific RNA-seq data and the predicted secreted subcellular location, ZmRALF12 peptides were detected at the filiform apparatus from 18 HAP and with increasing amounts at 24 HAP at both, the filiform apparatus and the filiform apparatus adjacent regions. We have added this new result as **Fig. 3i-l** and the corresponding description in the revised manuscript that contains also the figure legend.

2) Knockout/down of ZmRALF12 to show that ROS does not then accumulate at the expected timepoint after pollination/fertilisation.

Response: As suggested, we have generated *zmralf12* mutants using the CRISPR-Cas9-mediated genome editing system. We obtained more than ten independent mutant alleles of *ZmRALF12* to study the role of ZmRALF12 in ROS induction during pSC degeneration. Mutant seedlings showed severe growth retardation in soil, leading to about one-third of seedling death. This phenotype was not observed in other transgenic lines and clearly correlated to *zmralf12*. Expression analysis showed that *ZmRALF12* is moderately expressed in maize roots and leaves (Zhou et al. 2024. *Plant Cell*, **36**: 1673-1696), indicating it may also be required for maize vegetative development. The surviving *zmralf12* mutants exhibited a dwarf phenotype, and only a few of them could produce small cobs containing a limited number of ovaries and kernels.

Supplementary Fig. 6: Generation of a loss-of-function *zmralf12*^{-/-} mutation after CRISPR-Cas9 editing. **a** Schematic diagram of protein structure of wild-type ZmRALF12 and the CRISPR-Cas9-edited *zmralf12*^{-/-} mutant. SP, signal peptide; C, conserved cysteine residues. Gray box indicates missense sequence due to fragment deletion. **b** Sequencing result of the mutation site of *ZmRALF12* in the *zmralf12*^{-/-} mutant. Red arrow indicates deleted base pairs. **c** *zmralf12* mutant plants showed severe growth retardation and a dwarf phenotype. **d** Gene expression analysis of maize *RALF* family members in root and leaf tissues. *ZmRALF12* has moderate expression in maize roots and leaves. Transcript levels are provided in transcripts per million (TPM). **e** *zmralf12* mutant plants produced small cobs with limited number of ovaries and kernels. Scale bars, 5 cm.

We investigated ROS induction and mitochondria localization in the persistent synergid of *zmralf12* mutants. DCFH-DA and Mito-Tracker Red double-staining showed that in *zmralf12* mutants, ROS are not induced at the mitochondria-enriched area in the persistent synergid cell at 20 HAP. We have added the new data as **Fig. 3m** and **Supplementary Fig. 6**.

We next studied whether the absence of mitochondrial ROS production in *zmralf12* mutants indeed affects persistent synergid degeneration. Due to the strong vegetative phenotype and the limited number of ovaries, statistical analysis of persistent synergid degeneration phenotype with *zmralf12* mutants was not applicable. We thus used an alternative approach by manipulating the persistent synergid environment to mimic the ROS deficiency phenotype of *zmralf12* mutants. Ovules were harvested at 16 HAP and cultivated in an optimized culture medium or medium supplied with ROS scavenger CuCl_2 (5 mM). Compared to the control, post-fertilization ROS scavenger application (mimicking a ZmRALF12 mutant) significantly delayed persistent synergid cell death/elimination, as illustrated in novel **Fig. 5a–c**. Collectively, these results show that fertilization-induced ZmRALF12 accumulation at the filiform apparatus correlated with mitochondrial ROS production in the persisting synergid cell, which crucially determines timely persistent synergid cell death and elimination.

A number of central conclusions from this paper are based primarily on data from protoplasts of synergid cells, where protoplasting may induce cell stress responses. How closely these results reflect what happens in planta is a little difficult to assess without follow-up in planta experiments showing e.g. RALF12 responses and mitochondria localisation in planta. In experiments looking at localisation of ROS and mitochondria in dissected persistent synergids, it's not possible for an untrained eye to see where the filiform apparatus is located. Co-localisation (line 280) should therefore be demonstrated by experiments that show ROS or mitochondria location alongside a known reporter for the filiform apparatus.

Response: We thank the reviewer for these insightful suggestions. For manuscript revision, we now developed an optimized ovule cultivation system to further investigate mitochondria localization and ZmRALF12 response in ovule tissue and not in protoplasts (novel **Fig. 3e, f** and **Fig. 4a**). The results are consistent with the study using manually dissected synergid protoplasts (**Supplementary Fig. 7h**). According to this comment and also the suggestion from Reviewer #3, the *pZmSC3::ZmSC3-GFP* line was used in combination with Mito-Tracker Red staining to trace the filiform apparatus and its adjacent region during manual synergid cell isolation (**Supplementary Fig. 7d–g**). We found that mitochondria constitutively accumulate alongside the filiform apparatus adjacent region (FAAR) in synergid cells. The mitochondrial-enriched FAAR is maintained in synergid protoplasts after manual isolation, which exhibits efficient ZmRALF12 response and mitochondrial ROS accumulation shortly after ZmRALF12 peptide application. This is consistent with the ZmRALF12 response in ovule tissue (Fig. 4a and Supplementary Fig. 6a–c, h). We have added the new data in **Fig. 3**, **Fig. 4**, and **Supplementary Fig. 7**, and also added paragraphs to the manuscript discussing this data.

I found the section on ethylene responses vs ethylene production a little hard to follow. The authors hypothesised that increased ethylene production might trigger cell death, but then tested responses to exogenous ethylene. While changing ethylene production levels in maize synergids may be difficult, this would be the logical experiment given the stated hypothesis. Alternatively, some rewording of this section may be sufficient to make it easier to follow the logic.

Response: We are sorry that this section was difficult to follow and have now added sentences before the description of ethylene treatment experiments stating that: “In *Arabidopsis*,

microinjection of the ethylene precursor ACC into the female gametophyte led to premature synergid disintegration indicating that synergid cells appear to be more sensitive to ACC treatment and ethylene signaling (Völz et al. 2013. *Dev. Cell*, **25**: 310-316). However, whether ethylene itself plays a key role in synergid degeneration or components of the downstream signaling pathway is controversial, as ethylene production is not necessarily required for the death of synergid cells (Li et al. 2022. *Mol. Plant*, **15**: 354-362). To determine the effect of post-fertilization ethylene production on female gametophytic cells, we therefore performed ethylene application experiments with mature virgin maize ovules to mimic an ethylene burst environment after fertilization". This is also highlighted in the revised manuscript.

Minor points:

1. In the introduction, further explanation of the steps mentioned on line 56 would be helpful for a non-specialist reader.

Response: We thank the reviewer for this suggestion. We have now introduced descriptions for the three dPCD phases to make the content clearer and easier to understand. We write: "Despite its abundance and importance for plant life, the gene regulatory networks (GRNs) controlling the (i) induction (i.e., cell receiving signals that trigger the apoptotic pathway), (ii) effector (i.e., activation of lytic enzymes for cellular components cleavage), and (iii) degradation (i.e., cell disintegration and elimination) phases of dPCD are still poorly understood."

2. Addition of a timelapse video or video showing what happens with the synergids after pollen tube reception would be a very useful addition for non-specialists.

Response: We agree that it would greatly enhance clarity if we included a time-lapse video showing synergid behavior during double fertilization. However, unlike in *Arabidopsis*, live-cell imaging of female gametic cells in semi-in vivo cultivated ovules is to our knowledge not yet established in maize in any lab. Maize ovules are huge and contain many cell layers, which prevents live-cell imaging.

3. The words used to describe what happens to the persistent synergid vary, which would make the manuscript less accessible to some readers: elimination, removal, deletion, apoptosis, degeneration, programmed cell death. I think it would be better to restrict this to one or two terms which are clearly defined early in the manuscript and consistently used throughout.

Response: We think that a text reads more exciting if one uses different terms for the same meaning (here e.g. elimination, removal, deletion) to avoid redundancies. The senior author was criticized during a scientific rhetoric course as he used the word "indicate" in three successive sentences and was asked to use additional terms like "suggest". We would like to keep it here, because we think a text reads more attractive this way. Thank you for your understanding and guidance in enhancing the accessibility and consistency of our presentation.

4. Receptive synergid, degenerated receptive synergid, and degenerated synergid seem to be used as synonyms. It might be clearer to explain how these terms are used in the introduction and ensure consistent use thereafter.

Response: we are sorry for any unclarity – the terms are not used as synonyms and strongly depend on the context, which explains the meaning. A "receptive synergid" is the synergid that received the pollen tube; after pollen tube reception it becomes the "degenerated receptive synergid". We agree that the term "degenerated synergid" is unclear – it could be the "degenerated receptive synergid" or the "degenerated persisting synergid" – we removed the term "degenerated synergid" from the manuscript and use the more precise descriptions.

5. Line 54 – The authors mention that whether the persistent synergid undergoes PCD is disputed. I feel this could have been more deeply discussed in the conclusion section.

Response: We fully agree with the reviewer's point. We have discussed this now more thoroughly in the Discussion section of the revised manuscript and write: "PCD is an essential physiological process that plays a vital role in the development of both animals and plants. Cells undergoing PCD display characteristic morphological and biochemical features that include shrinkage, chromatin condensation, and activation of lytic enzymes for cellular component degradation. Those degenerative alterations were not observed in the persistent synergid cell except for nuclear disintegration. Due to a lack of cytological and molecular evidence, it has recently been discussed that the elimination of persistent synergid is not achieved by PCD, but by a cell fusion event followed by its nuclear degradation in the primary endosperm (Xie et al. 2022. *Curr. Opin. Plant Biol.*, **69**: 102271)."

6. Legend of Figure S1 could benefit from mention of the B73 maize line.

Response: We now added "Observation was performed with maize B73 inbred line" to the legend of Figure S1 in the revised manuscript.

7. Line 105 – typo. Should be 'collected' rather than 'collect'.

Response: We are sorry for our carelessness and have corrected the typo in the revised version.

8. How many cells were combined for each biological replicate of RNA-seq? Or is this scRNA-seq?

Response: Thanks for the helpful questions. In this study, we performed RNA-seq with manually dissected synergid cell samples at successive stages during persistent synergid PCD. For each biological replicate at each stage, 25–28 cells were used for mRNA isolation and cDNA preparation. We have added this information to the Methods section of the revised manuscript.

9. Line 123-125 – the authors state that ZmES4 goes down after fertilisation since it is required for pollen tube burst. However, it then goes up again from pSC18. Why might this occur?

Response: we agree that this might not be a good explanation and it is difficult to explain. We cannot exclude that it might also be required for persistent synergid cell disintegration, that transcript is first degraded and then *de novo* generated, etc.; we now write that "ZmES4 transcript levels remain high after fertilization which correlated well with the ZmES4-GFP reporter activity shown in Fig. 1a-c."

10. Where does the egg cell/zygote RNA-seq data come from? Was it also generated as part of this study? Either way, the source of this comparative data could be clearer in the manuscript.

Response: We are sorry that this was not clearly described. The egg cell/zygote RNA-seq data was derived from our previous study (Chen et al. 2017. *Plant Cell*, **29**: 2106-2125). This information was provided in the legend of Supplementary Data 4 of the previous manuscript and thus a bit hidden. We have now specified this more clearly in the main text as well as in the legends for **Supplementary Data 3** and **4** of the revised manuscript.

11. Figure 1 – I'm not convinced that 'High-precision' is needed or correct in the figure title. I think a more informative phrase could be used here, or the words simply omitted.

Response: Thank you for your comments. We noticed that "High-precision" could be confusing. In the revised manuscript, we have changed "High-precision transcriptome changes" to "Temporal resolution of transcriptome changes" to make it more clear.

12. Figure 1 j and k – Use of different arrow types to indicate the pollen tube entrance and sperm cell release regions would be better.

Response: We thank the reviewer for this suggestion. We have now used arrows and arrowheads to indicate pollen tube entrance and sperm cell release regions, respectively.

13. Figure 1m – some stats here to show significant differences between time points would be appropriate.

Response: We agree. The cell- and stage-specific RNA-seq data has been statistically analyzed with DESeq2 (Love et al. 2014. *Genome Biol.*, **15**: 1-21) and corrected for multiple testing. Information about statistical comparisons between cell stages was added to all figure panels depicting FPKM for specific genes according to adjusted *P*-values, as modified in the revised manuscript.

14. In Figure S2, the correlation between RNA-seq samples looks to be lower than the average R-squared given in the main text of 0.93. Obviously self-correlations should be excluded from this calculation, and 11 of the 66 values lie between 0.93 and 0.959 while 55 are between 0.769 and 0.93.

Response: During RNA-seq data analysis, the correlation coefficients between different samples were calculated according to Pearson's correlation coefficient method (**Supplementary Fig. 2b**). Here, we can see the correlation between three biological replicates and the variation between different stages. To demonstrate the expression values between each of the three biological replicates are highly correlated, we compared the R^2 between three biological replicates within the same stage and calculated the average R^2 value.

SC: 0.953 (SC_2 vs SC_1), 0.954 (SC_3 vs SC_1), 0.956 (SC_3 vs SC_2);

pSC12: 0.959 (pSC12_2 vs pSC12_1), 0.939 (pSC12_3 vs pSC12_1), 0.941 (pSC12_3 vs pSC12_2);

pSC18: 0.946 (pSC18_2 vs pSC18_1), 0.887 (pSC18_3 vs pSC18_1), 0.889 (pSC18_3 vs pSC18_2);

pSC24: 0.928 (pSC24_2 vs pSC24_1), 0.911 (pSC24_3 vs pSC24_1), 0.915 (pSC24_3 vs pSC24_2).

In order to make it clearer, we have now specified this in the main text of the revised manuscript and write "Comparison of the three biological replicates within the same stage showed that the expression values between them were highly correlated (average $R^2 = 0.93$)."

15. Line 145 – need to specify what at which time point you analysed gene expression patterns.

Response: We thank the reviewer for pointing out this issue. We have specified the time point accordingly to make it clearer and write "Next, we analyzed gene expression patterns in synergid cells of maize before pollination."

16. Line 149 and/or supplementary data 3 – need to explain what is meant by TOP200 category.

Response: We now provide a clear explanation of the TOP200 category. The corresponding legend for **Supplementary Data 3** was also improved.

17. Line 169 – typo. Should be 'which' rather than 'with'

Response: Corrected according to the suggestion.

18. Figure 2f, 3e, 3f, 5a – statistics needed.

Response: Information about statistical comparisons between cell stages was added now according to the adjusted *P*-values.

19. Figure 3h – inclusion of a negative control is needed

Response: A 0 min negative control has been included now as shown in **Supplementary Fig. 9** of the revised manuscript.

20. Line 818 – fluorescence intensity relative to what?

Response: We have also included the 0 min control in the quantification of ROS fluorescence intensity, to better compare the fluorescence intensity relative to the 0 min control and also between successive time points. Please see the detailed modifications in **Supplementary Fig. 9**.

21. Line 232-4 and Line 255– grammar needs fixing.

Response: The corresponding sentences have been corrected.

22. Figure 6 i, j, and l – statistics needed. Also, it would be better to show individual data points and keep presentation of expression data consistent between figures (also Fig S8).

Response: Statistical comparison information was added to the figure panels according to the adjusted *P*-values. In **Fig. 6i, j, l** and **Supplementary Fig. 8 (Supplementary Fig. 12** in the revised version), we used line charts instead of histograms in order to better exhibit an apparent increasing tendency after 18 HAP. In addition, the line charts in **Fig. 6i, j, and l** is consistent with **Fig. 6m** within this figure. Thank you for your understanding and guidance in enhancing the consistency of our presentation.

23. Line 323 – typo. Should be ‘but’ rather than ‘bot’.

Response: Corrected accordingly.

24. Line 387-9 – phrasing could be improved.

Response: We have rephrased the wording in the revised manuscript.

25. Line 427 – grammar

Response: The corresponding sentence has been corrected.

26. Line 432 – it would be helpful to refer to the figure which demonstrates the point you make here.

Response: The reference figure information has been added in the revised version.

27. Line 454 – spelling.

Response: The word “conserved” has been corrected accordingly.

28. Line 507 – should include *Agrobacterium* species and strain.

Response: We have now added the following information to the corresponding description: “The constructs were introduced into the *Agrobacterium tumefaciens* strain GV3101.”

29. Figure S9c and d – should the asterisk be lower down? Looks to be on the dSC based on other figures, rather than on the remnants of the pSC.

Response: To enhance clarity, we have now marked the specific cells in this figure and have modified the legend accordingly and write: “Secreted SC1-GFP signals (asterisk) are only visible at the residual filiform apparatus at the eliminated persistent synergid cell region at 26 HAP.”

30. Figure S5c – should include number of replicates

Response: The information about the number of replicates has now been included in the revised version.

31. Given you propose that PCD precedes fusion of the persistent synergid with the endosperm, does that imply that a dead cell fuses with a living cell? How does the endosperm recover from partial plasmolysis? Could expression of some previously uncharacterised genes during PCD in the synergid be to limit the effects of PCD on the endosperm after fusion? Perhaps some speculation on this could be included in the discussion.

Response: These are interesting questions. Our data showed that transcripts of PCD inducers, transcription factors, executors, as well as autophagy activities are highly activated shortly before SE fusion. Thus, PCD is well under way, but we think that the persistent synergid cell shortly before SE fusion is disintegrating but not a dead cell. We have added this information in the Discussion section of the revised manuscript and speculate that PCD executors are highly diluted in the huge central cells and that other endosperm factors probably limit the effects of synergid cell derived PCD factors thereby preventing its death.

We apologize for the confusion in presenting endosperm plasmolysis. In order to clearly visualize whether the persistent synergid cell is absent at 28 HAP, the embryo sacs were incubated in 12% mannitol solution for several minutes. This makes the primary endosperm exhibit partial plasmolysis at the micropylar end of the embryo sac. Therefore, we could see each cell clearly within this region. In fact, the endosperm does not undergo plasmolysis in physiological conditions in ovules. We have now specified this in the main text of the revised manuscript to make it clearer.

32. Supplementary data 9 – what is meant by ‘de novo’ here? The normal meaning of the phrase doesn’t seem quite appropriate here as many of the genes listed do have a low expression level prior to PCD.

Response: RNA-seq detects ‘background noise’ and therefore – like in other publications – we consider FPKM values <1 as not expressed. For clarity, we have now added the following phrase after ‘de novo’: “(<1 FPKM in synergid cells before fertilization)”.

33. Many genes are referred to as being ‘synergid-specific’ when either they are also expressed elsewhere according to the provided supplementary data files, and/or there is insufficient data to conclude that these genes are ONLY expressed in the synergids and not anywhere else in the plant. Maybe ‘synergid-expressed’ or ‘synergid-enriched’ would be more accurate as a descriptor?

Response: Among the >15,000 genes expressed in synergid cells, we identified only a very small number of genes that we consider as specific/predominant; these include 26 genes encoding secreted peptides and cell wall modifiers as well as 5 transcriptional regulators. We provide information on databases (such as UniProt) and papers (RNA-seq data by this study, by Chen et al. 2017 and by Yi et al. 2019) that served to identify specific/predominant genes. Noting that a subset of the genes also shows lower expression in egg cells, we now use the term “synergid-specific/predominant” instead of “synergid-specific”. To better substantiate their synergid-specific/predominant expression patterns, we have added information about their expression levels (FPKM values) in additional 22 samples during maize development (Wally et al. 2016. *Science*, **353**: 814-818). Please see the detailed information in Supplementary Data 4 and 6 of the revised manuscript.

34. Read order/labelling according to citation order could be improved in some of the figures.

Response: We have improved the citation order in the revised manuscript.

35. Supplementary data figures should be cited in strict numerical order.

Response: We have made modifications according to this helpful suggestion.

Reviewer #3 (Remarks to the Author):

Summary

This study used isolated synergid protoplasts from maize ovules at different stages post-fertilization to investigate the regulators of persistent synergid degradation. An accumulation of ROS was observed in persistent synergids 20 hours after pollination. RNA-Seq data indicated upregulation of a RALF signaling pathway and two NADPH Oxidases. Exogenous application of RALF 12, but not RALF1 or 2, to synergids induced ROS accumulation suggesting this peptide might regulate ROS accumulation during persistent synergid degradation. Mitochondrial association with the filiform apparatus was observed along with autophagosome accumulation 24 hours after pollination. Ethylene treatment could induce synergid degradation in ovules without affecting the egg or central cell, suggesting that synergids are primed for ethylene-mediated degradation. This study also identified new transcription factors that function as regulators of programmed cell death and are upregulated during persistent synergid degradation.

Major Strengths

The authors were able to isolate synergids to use for RNA-Seq and various treatments at different stages post-fertilization. The transcript data for persistent synergids at various time points allows for a dissection of how transcriptional regulation of genes changes from fertilization through the degradation of the persistent synergid. The data in the paper are well-presented and follow a clear logic to dissect the possible regulation of persistent synergid elimination. The most exciting result was that RALF12 was identified as the RALF family member upregulated during synergid degradation. RALF12 was able to induce ROS accumulation in isolated synergids in vitro. Previous studies investigating the role of ethylene in synergid degradation were done in Arabidopsis. The authors used exogenous application of ethylene to show that ethylene was sufficient to induce synergid degradation in maize ovules in vitro and that ethylene related genes were upregulated 24 hours after pollination, suggesting it might play a role in maize synergid degradation. 6 NAC transcription factors previously shown to induce programmed cell death are upregulated in synergids 24 hours after pollination and are candidate regulators of persistent synergid degradation. Overall, the manuscript makes and tests some interesting hypotheses about how the persistent synergid degrades following double fertilization.

Major Weaknesses

The authors didn't investigate whether making synergids protoplasts in order to isolate them affects the functionality of these cells. In most of the images of isolated synergid protoplasts, it appears that the synergids become round and the filiform apparatus does not keep its shape. For example, in Fig 3, some of the isolated synergids are completely round and it is difficult to see something that looks like a filiform apparatus. The authors should include information on how the filiform apparatus is identified in the protoplasts since many of their conclusions are based on identifying higher signal at the presumed filiform apparatus. Analysis of synergid protoplasts made from the pSC3::SC3 line (or another filiform apparatus-specific transgene) at the different time points would help define what happens to the filiform apparatus in protoplasts and could help refine the conclusions from the microscopy experiments.

Response: We appreciate the thoughtful review of our manuscript and the constructive suggestions. In the revised manuscript, we have added new data showing the consistency of mitochondria localization and RALF12 response both, in ovule tissue and in manually dissected synergid protoplasts (**Fig. 3e, f, Fig. 4a, and Supplementary Fig. 7a–c, h**). As suggested, the *pZmSC3::ZmSC3-GFP* line was used in combination with Mito-Tracker Red staining to mark the filiform apparatus and the stable mitochondrial-accumulating filiform apparatus adjacent region (FAAR) during manual synergid cell isolation (**Supplementary Fig. 7d–g**; please see also response #3 to Reviewer #2). Noting that the mitochondria-enriched region was detached from the thickened cell wall of the filiform apparatus during microdissection, we have accordingly described this area as the filiform apparatus adjacent region (FAAR). We thank both reviewers for pointing this out as we think the new data helped to significantly strengthen the manuscript.

Line 435 suggests that RALF12 specifically triggers mitochondrial ROS production in the filiform apparatus. However, the data presented in this paper does not fully support that model, especially since, based on many studies in Arabidopsis, the RALF peptides are expected to increase ROS through FERL-RLK signaling that upregulates the activity of plasma membrane associated RBOHs. Mitochondrial ROS is from a different source. The authors should discuss how the two NADPH Oxidases (RBOH3 and RBOH4) that are upregulated in the persistent synergid and presumably plasma-membrane localized might be involved in the mitochondrial ROS that accumulates in persistent synergids during the death process.

Response: We appreciate the reviewer for bringing up this point and decided to not only discuss the issue, but also to experimentally study mitochondrial ROS production in more detail (as suggested also below). We reported in the first version of the manuscript that fertilization-induced autocrine signal ZmRALF12 specifically triggers mitochondrial ROS production prominently at the mitochondria-enriched filiform apparatus adjacent region (FAAR) to mediate oxidative stress and PCD of the persistent synergid cell. Prompted by this

reviewer's comments, we investigated the underlying mechanisms of RALF-triggered mitochondria ROS production and accumulation.

We now show that preincubation with VAS2870, a specific inhibitor of NADPH oxidases, abolished ROS production in both, mitochondria and cytosol. This strongly supports the key role of RBOHs in promoting mitochondrial ROS production, being consistent with a recent study showing that the RBOH machinery acts in H₂O₂ accumulation in guard cell mitochondria to facilitate stomatal closure (Postiglione and Muday 2023. *Plant Physiol.*, **192**: 469-487). Previous studies in animal systems have shown that ROS production and signaling are intimately associated with Ca²⁺ channel activation, enabling Ca²⁺ flux into the cytosol or a particular subcellular compartment (e.g., mitochondria). Elevated Ca²⁺ uptake in mitochondria is strongly linked to respiratory chain activity stimulation, leading to higher amounts of mitochondrial ROS production (Görlach et al. 2015. *Redox Biol.*, **6**: 260-271). In support of this scenario, we find that the main product of RBOH activity, H₂O₂, effectively induces ROS accumulation in synergid mitochondria. Furthermore, both ZmRALF12-triggered RBOH-dependent and H₂O₂-induced mitochondrial ROS accumulation is Ca²⁺ channel-dependent. Collectively, these results support the picture (i) ZmRALF12 interacts with co-expressed ZmFERL1, ZmFERL16, and ZmLLG3 receptors (Supplementary Fig. 5), which upregulates RBOHs activities to generate ROS. (ii) ROS activates Ca²⁺ channels leading to an increase of Ca²⁺ release to the mitochondria, which induces ROS generation in the respiratory chain. (iii) Substantial downregulation of mitochondrial antioxidant enzymes after fertilization further promotes mitochondrial oxidative stress and damage (Fig. 4) leading to PCD. We have added this new result as **Supplementary Fig. 8** and added the corresponding description in the Results and Discussion sections of the revised manuscript.

Furthermore, it is unclear whether mitochondria associate with the filiform apparatus constitutively or if that is in fact induced by RALF12 and how the mitochondria contribute to the increased ROS. A 0HAP control should be included in Fig. 4 to show mitochondrial localization prior to fertilization or RALF12 treatment in both intact synergids and in synergid protoplasts

Response: We have now added the 0 HAP controls showing mitochondria distribution patterns in both female gametophytes containing intact synergids (**Fig. 3e**) and in synergid protoplasts (**Supplementary Fig. 7f, g**). Notably, we found that mitochondria accumulate at the FAAR constitutively already before fertilization and before RALF12 application. This polarized mitochondrial localization pattern may greatly facilitate efficient RALF12-induced mitochondrial ROS production and accumulation initially induced at this region.

This paper makes conclusions about the processes regulating the death of the persistent synergid from the RNA-Seq data. Many of the claims such as Ca²⁺ causing a shift from redox balance to oxidative stress were not proven experimentally. It's possible that post-transcriptional or translational regulation might show that protein activity doesn't correlate with the increased transcript levels of certain genes. In light of this potential complication, the authors should be more cautious in their interpretation of this transcriptome data.

Response: See also 2nd response above. We have carried out suggested experiments according to the insightful comments from the reviewers and carefully revised the whole text. Previous studies in animal systems demonstrated that the switch from redox balance to oxidative stress is largely mediated through Ca²⁺ signaling (Ermak and Davies 2002. *Mol. Immunol.*, **38**: 713-721). Being consistent with an oxidative stress response status in the persistent synergid cell at 24 HAP (Fig. 4), we do identify significant upregulation of genes encoding a plasma membrane Ca²⁺ permeable stress-gated cation channel (*ZmCSC1*), an ER membrane Ca²⁺ load-activated Ca²⁺ channel (*ZmTMCO1*), and a mitochondrial inner membrane Ca²⁺ uniporter (*ZmMCU2*) functioning in mediating Ca²⁺ uptake into mitochondria and activation of the cell death pathway (**Supplementary Fig. 11**). These suggest that like in animals, Ca²⁺ signaling very likely functions in mediating oxidative stress also in flowering plants. To clarify whether Ca²⁺ influx indeed plays an important role in this process, we have provided experimental data showing that RALF12-triggered mitochondrial ROS production

and accumulation is Ca²⁺ channel-dependent (**Supplementary Fig. 8**) and added the corresponding description in the Results section.

Lines 313-317 suggest that RALF12 triggers mitochondrial ROS production and leads to programmed cell death and autophagy. The current data presented only shows a link between ROS and autophagy through upregulated genes, but does not show a direct correlation. Imaging with MDC staining in RALF12 treated and untreated synergids to determine if the ROS induced by RALF12 can also induce autophagosome production would more directly show whether programmed cell death is triggered by RALF12.

Response: As suggested, we have performed the RALF12 treatment experiments using an optimized ovule cultivation system. We found that RALF12, but not RALF2, can effectively trigger autophagosome production and vacuole accumulation process in synergid cells, comparable to persistent synergid cells at 24 HAP (**Fig. 4h–k**). To better substantiate the conclusion, we have generated CRISPR-Cas9-edited loss-of-function *zmralf12* mutants and found that ROS are not induced in the persistent synergid cell at 20 HAP (**Fig. 3m** and **Supplementary Fig. 6**). Together these new data clearly show a direct correlation between RALF12-triggered mitochondrial ROS accumulation and oxidative stress-mediated PCD.

Minor Weaknesses

Statistical comparison for Figure 2F is missing. In the text (line 186) it suggests that all the transcription factors included decrease in the pSC compared to the control, but there isn't a statistical test done to demonstrate this and not all the genes look like they have reduced expression in the pSC. Nearly all figure panels depicting FPKM for specific genes in SC, pSC12, pSC18, and pSC24 are missing statistical comparisons (Figures 1M, 3E, 3F, 5A, supplemental figures 6, 7, 8).

Response: We apologize for this negligence. The expression patterns of the most abundant TFs have been specified more clearly in the revised text (Lines 200–203). Information about statistical comparisons between cell stages was now added to all figure panels depicting FPKM for specific genes according to adjusted *P*-values (please also refer to responses above to Reviewer #2).

In Figure 3H and supplemental figure 5 the synergid cells start looking deformed at 30 minutes or a concentration of 2µm RALF12. Does the deformation of the synergid have any biological relevance? Could the same dosage/time series depicted in supplemental figure 5 be done with RALF1 or RALF2 to dissect whether it is the ROS induced by RALF12 that causes this change in morphology or if the presence of a RALF in the absence of ROS would also cause the same change.

Response: The deformation at 30 minutes or at a concentration of 2µm RALF12 is due to the existence of a scarce amount of cell wall residuals during manual synergid microdissection. In some cases, the isolated synergid cells partially kept their original shape shortly after dissection (Supplementary Fig. 7f) and became round after incubation (Supplementary Fig. 7g). Usually, we directly acquire round-shaped cell-wall-less synergid protoplasts. This is also the case after RALF1 or RALF2 application (Fig. 4k and supplementary Fig. 7h). Based on the reviewer's questions/comments, we have named them now as pre-protoplasts (PP, non-round) and protoplasts (P, round), respectively, to enhance the clarity of our presentation.

Reviewer #4 (Remarks to the Author):

Response: we thank the ECR for helpful and constructive suggestions to further improve the manuscript.

Reviewer #5 (Remarks to the Author):

I co-reviewed this manuscript with one of the reviewers who provided the listed reports. This is part of the Nature Communications initiative to facilitate training in peer review and to provide appropriate recognition for Early Career Researchers who co-review manuscripts. *Response:* we thank the ECR for helpful and constructive suggestions to further improve the manuscript.

Manuscript NCOMMS-24-12001A

Title: “Fertilization-induced synergid cell death by RALF12-triggered ROS production and ethylene signaling” by Chen et al.

Point-by-Point Response to the Reviewer’s Comments

Reviewer #1 (Remarks to the Author):

The revised manuscript 'Fertilization-induced synergid cell death by RALF12-triggered ROS production and ethylene signaling' is clearly improved over the earlier version. While the high-resolution RNA-seq data provides clear evidence for the activation of molecular programs characteristic for programmed cell death (PCD) in persistent synergids, PCD still is no appropriate term, as a hallmark of PCD is the disintegration of DNA and the nucleus (1-3). Accordingly, the authors should refrain from using this term like previous authors did (4).

Response: We thank the reviewer for finding the manuscript is clearly improved. Furthermore, we believe that we have provided substantial cytological and molecular evidence showing that persistent synergid cells indeed undergoes dPCD. Many publications consider disintegration of both synergids as developmental PCD (dPCD) as introduced also in our report. We wrote, however, in the Introduction that “It remained unclear whether removal of the persistent synergid cell is achieved by PCD⁴” as discussed in an opinion paper and now provide a clear answer. We cite a more recent review of the authors mentioned in (3) below. Reference (4) describes a three-step inactivation process of the persisting synergid in Arabidopsis including disintegration of the nucleus, which – according to the reviewer – is a hallmark of PCD. In maize, we now found (i) *de novo* transcription of DNA degrading enzymes *ZmENDO1/BFN1* (encoding an endonuclease) and *ZmNEN1/4* (encoding an exonuclease; Fig. 6j), which further substantiated by the observation of persistent synergid nucleus disintegration (Fig. 6c). In addition to (ii) disintegration of the nucleus, we show (iii) mitochondrial ROS and (iv) autophagy as further hallmarks of PCD as well as (v) up-regulation of many PCD markers. Therefore, we think dPCD is the appropriate term and one of the novelties of our report.

What (still) stands out negatively about this paper is its disregard for highly relevant existing literature and data. This is not only misleading for reviewers, it also takes credit away from others' work: While the authors now integrate the requested publications regarding RALF and ethylene signaling the manuscript does not provide/discuss information on the extensive work previously published on the role of MYB98 during synergid disintegration in Arabidopsis and the existing knowledge on differential gene regulation by MYB98 during this process. Currently the MS reads as if the role of MYB98 for synergid disintegration was first discovered by the Dresselhaus lab. This needs correction.

Response: This is again a new comment, and we are sorry that the reviewer obtained the impression that we take credit away from others' work – this was of course never intended. Our research report is very complex and touches many biological fields. We cite 70 papers and could easily cite 150 more. Regarding MYB98 we have cited the first report (Punwani et al. 2007) showing in Arabidopsis that MYB98 positively regulates a battery of synergid-expressed genes. For clarity, we have now added in lines 211+212 “...and was previously shown to positive regulate a battery of synergid-expressed genes in Arabidopsis²⁹...”. We do not claim in the report a role of MYB98 during synergid disintegration as described above by the reviewer.

A key contribution of this study lies in demonstrating the role of RALF12 in an autocrine signaling pathway that triggers intracellular (mitochondrial) ROS production, which is distinct from the previously described apoplastic ROS burst in receptive synergids. The additional in vivo experiments, including the generation of a *ralf12* CRISPR null mutant, effectively substantiate the significance of RALF12-induced mitochondrial ROS. Further, the authors highlight how ethylene signaling acts synergistically with this RALF12-dependent mechanism to drive the disintegration of the persistent synergid.

Response: Thank you.

In their revised version, the authors also clarify previously raised questions about the timeline of synergid disintegration.

Response: Thank you.

The authors have made an effort to address the reviewer's concern by adopting the term 'synergid-specific/predominant.' However, the use of both terms can affect readability and has been applied somewhat inconsistently (e.g., in line 200: 'the most strongly expressed synergid-specific TFs').

Response: We carefully read the manuscript again but could not find another case except the one mentioned in line 200. Here we refer to "the top five most strongly expressed synergid-specific TFs" because these five TFs are indeed expressed exclusively in synergid cells but not in egg cells, and are almost not detected in any other investigated tissues (Supplementary Data 6). We added the reference data information (Supplementary Data 6) now in line 200 for clarity.

1 Fernández-Lázaro, D., Sanz, B. & Seco-Calvo, J. Mechanisms of programmed cell death: structural and functional pathways. A narrative review. *Investigación Clínica* (2024). <https://doi.org/10.54817/ic.v65n2a09>

2 Ren, H. et al. Calcium Signaling in Plant Programmed Cell Death. *Cells* 10 (2021). <https://doi.org/10.3390/cells10051089>

3 Huysmans, M., Lema A, S., Coll, N. S. & Nowack, M. K. Dying two deaths — programmed cell death regulation in development and disease. *Current Opinion in Plant Biology* 35, 37-44 (2017). <https://doi.org/https://doi.org/10.1016/j.pbi.2016.11.005>

4 Maruyama, D. & Higashiyama, T. The end of temptation: the elimination of persistent synergid cell identity. *Current Opinion in Plant Biology* 34, 122-126 (2016). <https://doi.org/https://doi.org/10.1016/j.pbi.2016.10.011>

Reviewer #2 (Remarks to the Author):

This is a revised version of the manuscript by Chen and colleagues on maize synergid cell death being triggered by ZmRALF12. The authors have responded thoroughly to all three reviewers, substantially strengthening and revising the manuscript in response to our many suggestions. This has clearly been quite a lot of work for which the authors should be commended. I believe the manuscript is now appropriate for publication.

Response: We thank the reviewer for the constructive comments to improve the manuscript.

Reviewer #3 (Remarks to the Author):

The revised manuscript is greatly improved due to the addition of experiments that helped the authors test the hypotheses generated based on their RNAseq data from unpollinated synergids and persistent synergids at various time points after pollination. In particular, the ROS experiments helped define the role of ROS in the persistent synergid. The link between RboH- and mitochondrial generation of ROS is intriguing and provides a new model for how RALF signaling promotes programmed cell death. The additional RALF12 crispr and immunolocalization data also strengthened the conclusion that RALF12 is involved in regulating persistent synergid cell death. I'm satisfied with the authors' responses to my critiques on the first version of the manuscript. I think that they were also able to adequately respond to the other reviewers' concerns.

Minor suggestions:

Lines 534: Notably, we show that in consistence with XX" should be changed to "Consistent with XX,"

Line 563: "Moreover, in consistence with activation..." should be "Consistent with activation..."

Figure 2 legend: please note in the legend that the data in the first pie chart in panel a is synergids before pollination

Figure 4: The label in panel 4K should be ZmRALF2

Figure 6: Change the d-1 label so that it is all on the same line

Response: We are happy that we were able to satisfy the reviewer and have corrected all minor edits listed above.

Reviewer #5 (Remarks to the Author):

I co-reviewed this manuscript with one of the reviewers who provided the listed reports.

Response: We thank the ECR.